# A Meta-Transfer Objective for Learning to Disentangle Causal Mechanisms

**Yoshua Bengio**[1,2,5]    **Tristan Deleu**[1]    **Nasim Rahaman**[4]    **Nan Rosemary Ke**[3]
**Sébastien Lachapelle**[1]    **Olexa Bilaniuk**[1]    **Anirudh Goyal**[1]    **Christopher Pal**[3,5]
Mila – Montreal, Quebec, Canada

## Abstract

We propose to use a meta-learning objective that maximizes the speed of transfer on a modified distribution to learn how to modularize acquired knowledge and discover causal dependencies. In particular, we focus on how to factor a joint distribution into appropriate conditionals, consistent with the causal directions. To replace the assumption that the test cases are of the same distribution as the training examples, this method exploits the assumption that the changes in distributions are localized (e.g. to one of the marginals, for example due to an intervention on a cause). We prove that under this assumption of localized changes in causal mechanisms, the correct causal graph will tend to have only a few of its parameters with non-zero gradient, i.e. that need to be adapted (those of the modified variables). We argue and observe experimentally that this leads to faster adaptation, and use this property to define a meta-learning surrogate score which, in addition to a continuous parametrization of graphs, would favour correct causal graphs, making it possible to discover causal structure by gradient-based methods. Finally, motivated by the AI agent point of view (e.g. of a robot discovering its environment autonomously), we consider how the same objective can discover the causal variables themselves, as a transformation of observed low-level variables with no causal meaning. Experiments in the two-variable case validate the proposed ideas and theoretical results.

## 1   Introduction

The data used to train our models is often assumed to be independent and identically distributed (iid.), according to some unknown distribution. Likewise, the performance of a machine learning model is typically evaluated using test samples from the same distribution, assumed to be representative of the learned system's usage. While these assumptions are well analyzed from a statistical point of view, they are rarely satisfied in many real-world applications. For example, an accident on a major highway could completely perturb the trajectories of cars, and a driving policy trained in a static way might not be robust to such changes. Ideally, we would like our models to generalize well and *adapt quickly* to out-of-distribution data.

However, this comes at a price – in order to successfully transfer to a novel distribution, one might need additional information about these distributions. In this paper, we are not considering assumptions on the data distribution itself, but rather on how it changes (e.g., when going from a training distribution to a transfer distribution, possibly resulting from some agent's actions). We focus on the assumption that the changes are sparse when the knowledge is represented in an appropriately modularized way, with only one or a few of the modules having changed. This is especially relevant when the distributional change is due to actions by one or more agents, because agents intervene at a particular place and time, and this is reflected in the form of the interventions discussed in the causality literature (Pearl, 2009; Peters et al., 2016), where a single causal variable is clamped to a particular value or a random variable. In general, it is difficult for agents to influence many underlying causal variables at a time, and although this paper is not about agent learning as such, this is a property of the world that we propose to exploit here, to help discovering these variables

---

[1] Université de Montréal, [2] CIFAR Senior Fellow, [3] École Polytechnique Montréal, [4] Max-Planck Institute for Intelligent Systems, Tübingen, [5] Canada CIFAR AI Chair

and how they are causally related to each other. In this context, the causal graph is a powerful tool because it tells us how perturbations in the distribution of intervened variables will propagate to all other variables and affect their distributions.

As expected, it is often the case that the causal structure is not known in advance. The problem of causal discovery then entails obtaining the causal graph, a feat which is in general achievable only with strong assumptions. One such assumption is that a learner that has learned to capture the correct structure of the true underlying data-generating process should still generalize to the case where the structure has been perturbed in a certain, restrictive way. This can be illustrated by considering the example of temperature and altitude from Peters et al. (2017): a learner that has learned to capture the mechanisms of atmospheric physics by learning that it makes more sense to predict temperature from the altitude (rather than vice versa) given training data from (say) Switzerland, will still remain valid when tested on out-of-distribution data from a less mountainous country like (say) the Netherlands. It has therefore been suggested that the out-of-distribution robustness of predictive models can be used to guide the inference of the true causal structure (Peters et al., 2016; 2017).

How can we exploit the assumption of localized change? As we explain theoretically and verify experimentally here, if we have the right knowledge representation, then we should get fast adaptation to the transfer distribution when starting from a model that is well trained on the training distribution. This arises because of our assumption that the ground truth data generative process is obtained as the composition of independent mechanisms, and that very few ground truth mechanisms and parameters need to change when going from the training distribution to the transfer distribution. A model capturing a corresponding factorization of knowledge would thus require just a few updates, a few examples, for this adaptation to the transfer distribution. As shown below, the expected gradient on the unchanged parameters would be near 0 (if the model was already well trained on the training distribution), so the effective search space during adaptation to the transfer distribution would be greatly reduced, which tends to produce faster adaptation, as found experimentally. Thus, based on the assumption of small change in the right knowledge representation space, we can define a meta-learning objective that measures the speed of online adaptation in order to optimize the way in which knowledge should be represented, factorized and structured. This is the core idea presented in this paper.

Returning to the example of temperature and altitude: when presented with out-of-distribution data from the Netherlands, we expect the correct model to adapt faster given a few *transfer samples* of actual weather data collected in the Netherlands. Analogous to the case of robustness, the adaptation speed can then be used to guide the inference of the true causal structure of the problem at hand, possibly along with other sources of signal about causal structure.

**Contributions.** We first verify on synthetic data that the model that correctly captures the underlying causal structure adapts faster when presented with data sampled after a performing certain interventions on the true two-variable causal graph (which is unknown to the learner). This suggests that the adaptation speed can indeed function as a score to assess how well the learner fits the underlying causal graph. We then use a smooth parameterization of the considered causal graph to directly optimize this score in an end-to-end gradient-based manner. Finally, we show in a simple setting that the score can be exploited to *disentangle* the correct causal variables given an unknown mixture of the said variables.

## 2 Which is Cause and Which is Effect?

As an illustrative example of the proposed ideas, let us consider two discrete random variables $A$ and $B$, each taking $N$ possible values. We assume that $A$ and $B$ are correlated, without any hidden confounder. Our goal is to determine whether the underlying causal graph is $A \rightarrow B$ ($A$ *causes* $B$), or $B \rightarrow A$. Note that this underlying causal graph cannot be identified from observational data from a single (training) distribution $p$ only, since both graphs are Markov equivalent for $p$ (Verma & Pearl, 1991); see Appendix A. In order to disambiguate between these two hypotheses, we will use samples from some transfer distribution $\tilde{p}$ in addition to our original samples from the training distribution $p$.

### 2.1 The advantage of the correct causal model

Without loss of generality, we can fix the true causal graph to be $A \rightarrow B$, which is unknown to the learner. Moreover, to make the case stronger, we will consider a setting called *covariate shift* (Rojas-Carulla et al., 2018; Quionero-Candela et al., 2009), where we assume that the change (again,

whose nature is unknown to the learner) between the training and transfer distributions occurs after an intervention on the cause $A$. In other words, the marginal of $A$ changes, while the conditional $p(B \mid A)$ does not, i.e. $p(B \mid A) = \tilde{p}(B \mid A)$. Changes on the cause will be most informative, since they will have direct effects on $B$. This is sufficient to fully identify the causal graph (Hauser & Bühlmann, 2012).

In order to demonstrate the advantage of choosing the causal model $A \to B$ over the anti-causal $B \to A$, we can compare how fast the two models can adapt to samples from the transfer distribution $\tilde{p}$. We quantify the speed of adaptation as the log-likelihood after multiple steps of fine-tuning via (stochastic) gradient ascent, starting with both models trained on a large amount of data from the training distribution. In Figure 1 (see Section 3.3 for the experimental setup), we can see that the model corresponding to the underlying causal model adapts faster. Moreover, the difference is more significant when adapting on a small amount of data, of the order of 10 to 30 samples from the transfer distribution. We will make use of this property as a noisy signal to infer the direction of causality, which here is equivalent to choosing how to modularize the joint distribution.

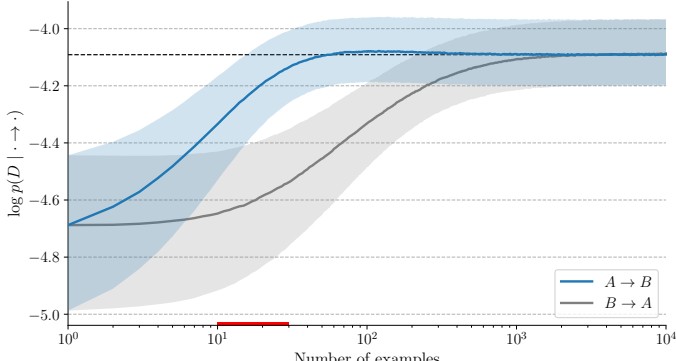

Figure 1: Adaptation to the transfer distribution (average log-likelihood of the model during fine-tuning adaptation to transfer examples, vertical axis), as more transfer examples are seen by the learner (horizontal axis). The curves are the median over 20,000 runs, with their 25-75th quantiles intervals. The dotted line is the asymptotic log-likelihood (here, that of the ground truth $\tilde{p}$). The red region corresponds to the range where the effect is the most significant (10-30 samples from the transfer distribution).

## 2.2 PARAMETER COUNTING ARGUMENT

A simple parameter counting argument can help us understand what we are observing in Figure 1. Since we are using gradient ascent for the adaptation, let's first inspect how the gradients of the log-likelihood wrt. each module behave under the transfer distribution.

**Proposition 1.** *Let $G$ be a causal graph, and $p$ a (training) distribution that factorizes according to $G$, with parameters $\theta$. Let $\tilde{p}$ be a second (transfer) distribution that also factorizes according to $G$. If the training and transfer distributions have the same conditional probability distributions for all $V_i$ but a subset $C$ (e.g. the transfer distribution is the result of an intervention on the nodes in $C$):*

$$p(V_i \mid \mathrm{Pa}_G(V_i)) \overset{d}{=} \tilde{p}(V_i \mid \mathrm{Pa}_G(V_i)) \qquad \forall V_i \notin C \tag{1}$$

*then the expected gradient w.r.t. the parameters $\theta_i$ such that $V_i \notin C$ of the log-likelihood under the transfer distribution will be zero*

$$\forall V_i \notin C, \ \mathbb{E}_{V \sim \tilde{p}} \left[ \frac{\partial \log p(V)}{\partial \theta_i} \right] = 0. \tag{2}$$

Proposition 1 (see proof in Appendix B.1) suggests that if both distributions factorize according to the correct causal graph, then only the parameters of the mechanisms that changed between the training and transfer distributions need to be updated. This effectively reduces the number of parameters that need to be adapted compared to any other factorization over a different graph. It also affects the number of examples necessary for the adaptation, since the sample complexity of a model grows approximately linearly with the VC-dimension (Ehrenfeucht et al., 1989; Vapnik & Chervonenkis, 1971), which itself also grows approximately linearly with the number of parameters (for linear

models and neural networks; Shalev-Shwartz & Ben-David, 2014). Therefore we argue that the performance on the transfer distribution (in terms of log-likelihood) will tend to improve faster if it factorizes according to the correct causal graph, an assertion which may not be true for every graph but that we can test by simulations.

Recall that in our example on two discrete random variables (each taking say $N$ values), we assumed that the underlying causal model is $A \to B$, and the transfer distribution is the result of an intervention on the cause $A$. If the model we learn on the training distribution factorizes according to the correct graph, then only $N - 1$ free parameters should be updated to adapt to the shifted distribution, accounting for the change in the marginal distribution $\tilde{p}(A)$, since the conditional $\tilde{p}(B \mid A) = p(B \mid A)$ stays invariant. On the other hand, if the model factorizes according to the anti-causal graph $B \to A$, then the parameters for both the marginal $\tilde{p}(B)$ and the conditional $\tilde{p}(A \mid B)$ must be adapted. Assuming there is a linear relationship between sample complexity and the number of free parameters, the sample complexity would be $O(N^2)$ for the anti-causal graph, compared to only $O(N)$ for the true underlying causal graph $A \to B$.

## 3 THE META-TRANSFER OBJECTIVE

Since the speed of adaptation to some transfer distribution is closely related to the right modularization of knowledge, we propose to use it as a noisy signal to iteratively improve inference of the causal structure from data. Moreover, we saw in Figure 1 that the gap between correct and incorrect models is largest with a small amount of transfer data. In order to compare how fast some models adapt to a change in distribution, we can quantify the speed of adaptation based on their accumulated online performance after fine-tuning with gradient ascent on few examples from the transfer distribution. More precisely, given a small "intervention" dataset $\mathcal{D}_{int} = \{\mathbf{x}_t\}_{t=1}^T$ from $\tilde{p}$, we can define the *online likelihood* as

$$\mathcal{L}_G(\mathcal{D}_{int}) = \prod_{t=1}^T p(\mathbf{x}_t \,;\, \theta_G^{(t)}, G) \qquad \begin{aligned} \theta_G^{(1)} &= \hat{\theta}_G^{ML}(\mathcal{D}_{obs}) \\ \theta_G^{(t+1)} &= \theta_G^{(t)} + \alpha \nabla_\theta \log p(\mathbf{x}_t \,;\, \theta_G^{(t)}, G), \end{aligned} \qquad (3)$$

where $\theta_G^{(t)}$ aggregates all the modules' parameters in $G$ after $t$ steps of fine-tuning with gradient ascent, with learning rate $\alpha$, starting from the maximum-likelihood estimate $\hat{\theta}_G^{ML}(\mathcal{D}_{obs})$ on a large amount of data $\mathcal{D}_{obs}$ from the training distribution $p$. Note that, in addition to its contribution to the update of the parameters, each data point $\mathbf{x}_t$ is also used to evaluate the performance of our model so far; this is called a *prequential analysis* (Dawid, 1984), also corresponding to sequential cross-validation (Gingras et al., 1999). From a structure learning perspective, the online likelihood (or, equivalently, its logarithm) can be interpreted as a score we would like to maximize, in order to recover the correct causal graph.

### 3.1 CONNECTION TO THE BAYESIAN SCORE

We can draw an interesting connection between the online log-likelihood, and a widely used score in structure learning called the *Bayesian score* (Heckerman et al., 1995; Geiger & Heckerman, 1994). The idea behind this score is to treat the problem of learning the structure from a fully Bayesian perspective. If we define a prior over graphs $p(G)$ and a prior $p(\theta_G \mid G)$ over the parameters of each graph $G$, the Bayesian score is defined as $\text{score}_B(G \,;\, \mathcal{D}_{int}) = \log p(\mathcal{D}_{int} \mid G) + \log p(G)$, where $p(\mathcal{D}_{int} \mid G)$ is the marginal likelihood

$$p(\mathcal{D}_{int} \mid G) = \prod_{t=1}^T p(\mathbf{x}_t \mid \mathbf{x}_1, \ldots, \mathbf{x}_{t-1}, G) = \prod_{t=1}^T \left[ \int_{\Theta_G} p(\mathbf{x}_t \mid \theta_G, G) p(\theta_G \mid \mathbf{x}_{1:t-1}, G) \, \mathrm{d}\theta_G \right]. \tag{4}$$

In the online likelihood, the adapted parameters $\theta_G^{(t)}$ act as a summary of past data $\mathbf{x}_{1:t-1}$. Eq. (3) can be seen as an approximation of the marginal likelihood in Eq. (4), where the posteriors over the parameters $p(\theta_G \mid \mathbf{x}_{1:t-1}, G)$ is approximated by the point estimate $\theta_G^{(t)}$. Therefore, the online log-likelihood provides a simple way to approximate the Bayesian score, which is often intractable.

### 3.2 A SMOOTH PARAMETRIZATION OF THE CAUSAL STRUCTURE

Due to the super-exponential number of possible Directed Acyclic Graphs (DAGs) over $n$ nodes, the problem of searching for a causal structure that maximizes some score is, in general, NP-hard

(Chickering, 2002a). However, we can parametrize our belief about causal graphs by keeping track of the probability for each directed edge to be present. This provides a smooth parametrization of graphs, which hinges on gradually changing our belief in individual binary decisions associated with each edge of the causal graph. This allows us to define a fully differentiable meta-learning objective, with all the beliefs being updated at the same time by gradient descent.

In this section, we study the simplest version of this idea, applied to our example on two random variables from Section 2. Recall that here, we only have two hypotheses to choose from: either $A \rightarrow B$ or $B \rightarrow A$. We represent our belief of having an edge connecting $A$ to $B$ with a structural parameter $\gamma$ such that $p(A \rightarrow B) = \sigma(\gamma)$, where $\sigma(\gamma) = 1/(1 + \exp(-\gamma))$ is the sigmoid function. We propose, as a *meta-transfer objective*, the negative log-likelihood $\mathcal{R}$ (a form of regret) over the mixture of these two models, where the mixture parameter is given by $\sigma(\gamma)$:

$$\mathcal{R}(\mathcal{D}_{int}) = -\log\left[\sigma(\gamma)\mathcal{L}_{A\rightarrow B}(\mathcal{D}_{int}) + (1 - \sigma(\gamma))\mathcal{L}_{B\rightarrow A}(\mathcal{D}_{int})\right] \quad (5)$$

This meta-learning mixture combines the online adaptation likelihoods of each model over one meta-example or episode (specified by a $\mathcal{D}_{int} \sim \tilde{p}$), rather than considering and linearly mixing the per-example likelihoods as in ordinary mixtures.

In the experiments below, after each episode involving $T$ examples $\mathcal{D}_{int}$ from the transfer distribution $\tilde{p}$, we update $\gamma$ by doing one step of gradient descent, to reduce the regret $\mathcal{R}$. Therefore, in order to update our belief about the edge $A \rightarrow B$, the quantity of interest is the gradient of the objective $\mathcal{R}$ with respect to the structural parameter, $\partial \mathcal{R}/\partial \gamma$. This gradient is pushing $\sigma(\gamma)$ towards the posterior probability that the correct model is $A \rightarrow B$, given the evidence from the transfer data:

**Proposition 2.** *The gradient of the negative log-likelihood of the transfer data $\mathcal{D}_{int}$ in Equation (5) wrt. the structural parameter $\gamma$ is given by*

$$\frac{\partial \mathcal{R}}{\partial \gamma} = p(A \rightarrow B) - p(A \rightarrow B \mid \mathcal{D}_{int}), \quad (6)$$

*where $p(A \rightarrow B \mid \mathcal{D}_{int})$ is the posterior probability of the hypothesis $A \rightarrow B$ (when the alternative is $B \rightarrow A$). Furthermore, this can be equivalently written as*

$$\frac{\partial \mathcal{R}}{\partial \gamma} = \sigma(\gamma) - \sigma(\gamma + \Delta), \quad (7)$$

*where $\Delta = \log \mathcal{L}_{A\rightarrow B}(\mathcal{D}_{int}) - \log \mathcal{L}_{B\rightarrow A}(\mathcal{D}_{int})$ is the difference between the online log-likelihoods of the two hypotheses on the transfer data $\mathcal{D}_{int}$.*

The proof is given in Appendix B.2. Note how the posterior probability is basically measuring which hypothesis is better explaining the transfer data $\mathcal{D}_{int}$ overall, along the adaptation trajectory. This posterior depends on the difference in online log-likelihoods $\Delta$, showing the close relation between minimizing the regret $\mathcal{R}$ and maximizing the online log-likelihood score. The sign and magnitude of $\Delta$ have a direct effect on the convergence of the meta-transfer objective. We can show that the meta-transfer objective is guaranteed to converge to one of the two hypotheses.

**Proposition 3.** *With stochastic gradient descent (and an appropriately decreasing learning rate) on $\mathbb{E}_{\mathcal{D}_{int}}[\mathcal{R}(\mathcal{D}_{int})]$, where the gradient steps are given by Proposition 2, the structural parameter converges towards*

$$\begin{aligned} \sigma(\gamma) \rightarrow 1 \quad &\text{if } \mathbb{E}_{\mathcal{D}_{int}}[\mathcal{L}_{A\rightarrow B}(\mathcal{D}_{int})] > \mathbb{E}_{\mathcal{D}_{int}}[\mathcal{L}_{B\rightarrow A}(\mathcal{D}_{int})] \\ \text{or } \sigma(\gamma) \rightarrow 0 \quad &\text{otherwise} \end{aligned} \quad (8)$$

This proposition (proved in Appendix B.3) shows that optimizing $\gamma$ is equivalent to picking the hypothesis that has the smallest regret (or fastest convergence), measured as the accumulated log-likelihood of the transfer dataset $\mathcal{D}_{int}$ during adaptation. The distribution over datasets $\mathcal{D}_{int}$ is similar to a distribution over tasks in meta-learning. This analogy with meta-learning also appears in our gradient-based adaptation procedure, which is linked to existing methods like the first-order approximation of MAML (Finn et al., 2017), and its related algorithms (Grant et al., 2018; Kim et al., 2018; Finn et al., 2018). The pseudo-code for the proposed algorithm is given in Algorithm 1.

This smooth parametrization of the causal graph, along with the definition of the meta-transfer objective in Equation (5), can be extended to graphs with more than 2 variables. This general formulation builds on the bivariate case, where decisions are binary for each individual edge of the graph. See Appendix E for details and a generalization of Proposition 2; the structure of Algorithm 1 remains unchanged. Experimentally, this generalization of the meta-transfer objective proved to be effective on larger graphs (Ke et al., 2019), in work following the initial release of this paper.

---

**Algorithm 1** Meta-learning algorithm for learning the structural parameter

---
**Require:** Two graph candidates $G = A \rightarrow B$ and $G = B \rightarrow A$
**Require:** A training distribution $p$ that factorizes over the correct causal graph
 1: Set the initial structural parameter $\gamma = 0$ ▷ equal belief for both hypotheses
 2: Sample a large dataset $\mathcal{D}_{obs}$ from the training distribution $p$
 3: Pretrain the parameters of both models with maximum likelihood on $\mathcal{D}_{obs}$
 4: **for each** episode **do**
 5:     Draw a transfer distribution $\tilde{p}$ (via an intervention)
 6:     Sample a (small) transfer dataset $\mathcal{D}_{int} = \{\mathbf{x}_t\}_{t=1}^T$ from $\tilde{p}$
 7:     **for** $t = 1, \ldots, T$ **do**
 8:         Accumulate the online log-likelihood for both models $\mathcal{L}_{A \rightarrow B}$ and $\mathcal{L}_{B \rightarrow A}$ as they adapt
 9:         Do one step of gradient ascent for both models: $\theta_G^{(t+1)} = \theta_G^{(t)} + \alpha \nabla_\theta \log p(\mathbf{x}_t \,;\, \theta_G^{(t)}, G)$
10:     Compute the regret $\mathcal{R}(\mathcal{D}_{int})$
11:     Compute the gradient of the regret wrt. $\gamma$ (see Proposition 2)
12:     Do one step of gradient descent on the regret w.r.t. $\gamma$
13:     Reset the models' parameters to the maximum likelihood estimate on $\mathcal{D}_{obs}$

---

## 3.3 EXPERIMENTAL RESULTS

To illustrate the convergence result from Proposition 3, we experiment with learning the structural parameter $\gamma$ in a bivariate model. Following the setting presented in Section 2.1, we assume in all our experiments that $A$ and $B$ are two correlated random variables, and the underlying causal model (unknown to the algorithm) is fixed to $A \rightarrow B$. Recall that both variables are observed, and there is no hidden confounding factor. Since the correct causal model is $A \rightarrow B$, the structural parameter should converge correctly, with $\sigma(\gamma) \rightarrow 1$. The details of the experimental setups, as well as details about the models, can be found in Appendix C.

We first experiment with the case where both $A$ and $B$ are discrete random variables, taking $N$ possible values. In this setting, we explored how two different parametrizations of the conditional probability distributions (CPDs) might influence the convergence of the structural parameter. In the first experiment, we parametrized the CPDs as multinomial logistic CPDs (Koller & Friedman, 2009), maintaining a tabular representation of the conditional probabilities. For example, the conditional distribution $p(B \mid A)$ is represented as

$$p(B = j \mid A = i \,;\, \theta) = \frac{\exp(\theta_{ij})}{\sum_k \exp(\theta_{ik})}, \tag{9}$$

where the parameter $\theta$ is an $N \times N$ matrix. We used a similar representation for the other marginal and conditional distributions $p(A)$, $p(B)$ and $p(A \mid B)$. In a second experiment, we used structured CPDs, parametrized with multi-layer perceptrons (MLPs) with a softmax nonlinearity at the output layer. The advantage over a tabular representation is the ability to share parameters for similar contexts, and reduces the overall number of parameters required for each module. This would be crucial if either the number of categories $N$, or the number of variables, increased significantly.

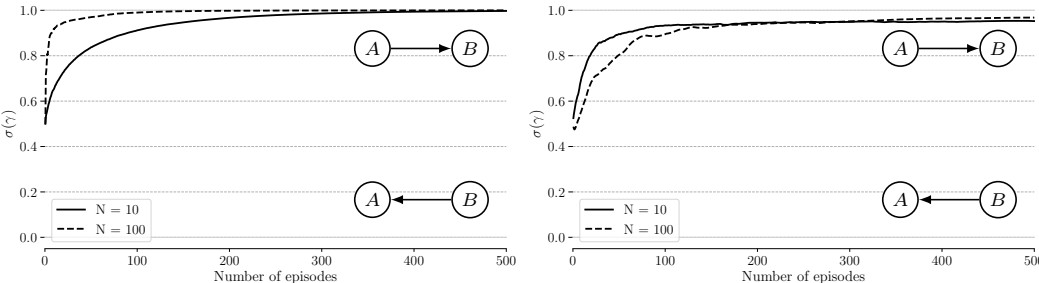

Figure 2: Evolution of the belief that $A \rightarrow B$ is the correct causal model, as the number of episodes increases, starting with an equal belief for both hypotheses. (Left) multinomial logistic CPDs, (right) MLP parametrization.

In Figure 2, we show the evolution of $\sigma(\gamma)$, which is the model's belief of $A \to B$ being the correct causal model, as the number of episodes increases, for different values of $N$. As expected, the structural parameter converges correctly to $\sigma(\gamma) \to 1$, within a few hundreds episodes. This observation is consistent in both experiments, regardless of the parametrization of the CPDs. Interestingly, the structural parameter tends to converge faster with a larger value of $N$ and a tabular representation, illustrating the effect of the parameter counting argument described in Section 2.2, which is stronger as $N$ increases. Precisely *when generalization is more difficult (too many parameters and too few examples), we get a stronger signal about the better modularization.*

We also experimented with $A$ and $B$ being continuous random variables, where they follow either multimodal distributions, or they are linear-Gaussian. Similar to Figure 2, we found that the structural parameter $\sigma(\gamma)$ consistently converges to the correct causal model as well. See Appendix C.3 and Appendix C.4 for details about these experiments.

## 4 REPRESENTATION LEARNING

So far, we have assumed that all the variables in the causal graph are fully observed. However, in many realistic scenarios for learning agents, the learner might only have access to low-level observations (e.g. sensory-level data, like pixels or acoustic samples), which are very unlikely to be individually meaningful as causal variables. In that case, our assumption that the changes in distributions are localized might not hold at this level of observed data. To tackle this, we propose to follow the deep learning objective of disentangling the underlying causal variables (Bengio et al., 2013), and learn a representation in which the variables can be meaningfully cause or effect of each other. Our approach is to jointly learn this representation, as well as the causal graph over the latent variables.

We consider the simplest setting where the learner maps raw observations to a hidden representation space with two causal variables, via an encoder $\mathcal{E}$. The encoder is trained such that this latent space helps to optimize the meta-transfer objective described in Section 3. We consider the parameters of the encoder, as well as $\gamma$ (see Section 3.2), as part of the set of structural meta-parameters to be optimized. We assume that we have two raw observed variables $(X, Y)$, generated from the true causal variables $(A, B)$ via the action of a ground truth decoder $\mathcal{D}$ (or generator network), that the learner is not aware of. This allows us to still have the ability to intervene on the underlying causal variables (e.g. to shift from training to transfer distributions) for the purpose of conducting experiments, while the learner only sees data from $(X, Y)$.

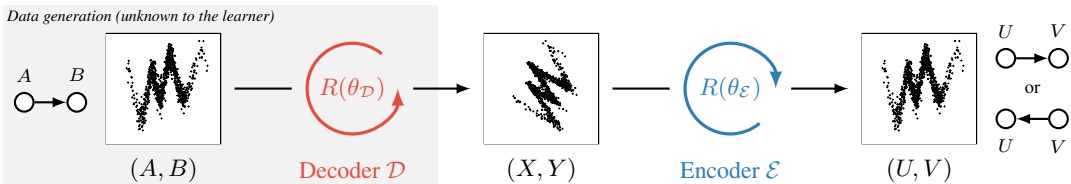

Figure 3: The complete experimental setup. The ground-truth variables $(A, B)$ are assumed to originate from the true underlying causal model, but the observations available to the learner are samples from $(X, Y)$. The observed variables $(X, Y)$ are derived from $(A, B)$ via the action of a decoder $\mathcal{D}$. The encoder $\mathcal{E}$ must be learned to undo this action of the decoder, and thereby recover the true causal variables up to symmetries. The components of the data generation on the left are hidden to the model.

In this experiment, we only want to validate the proposed meta-objective as a way to recover a good encoder, and we assume that both the decoder $\mathcal{D}$ and the encoder $\mathcal{E}$ are rotations, whose angles are $\theta_{\mathcal{D}}$ and $\theta_{\mathcal{E}}$ respectively. The encoder maps the raw observed variables $(X, Y)$ to the latent variables $(U, V)$, over which we want to infer the causal graph. Similar to our experiments in Section 3.3, we assume that the underlying causal graph is $A \to B$, and the transfer distribution $\tilde{p}$ (now over $(X, Y)$) is the result of an intervention over $A$. Therefore, the encoder should ideally recover the structure $U \to V$ in the learned latent space, along with the angle of the encoder $\theta_{\mathcal{E}} = -\theta_{\mathcal{D}}$. However, since the encoder is not uniquely defined, $V \to U$ might also be a valid solution, if the encoder is $\theta_{\mathcal{E}} = -\pi/2 - \theta_{\mathcal{D}}$. Details about the experimental setup are provided in Appendix D. In Figure 4,

we consider that the learner succeeds, since both structural parameters converge to one of the two options. This shows how minimizing the meta-transfer objective can disentangle (here in a very simple setting) the ground-truth variables.

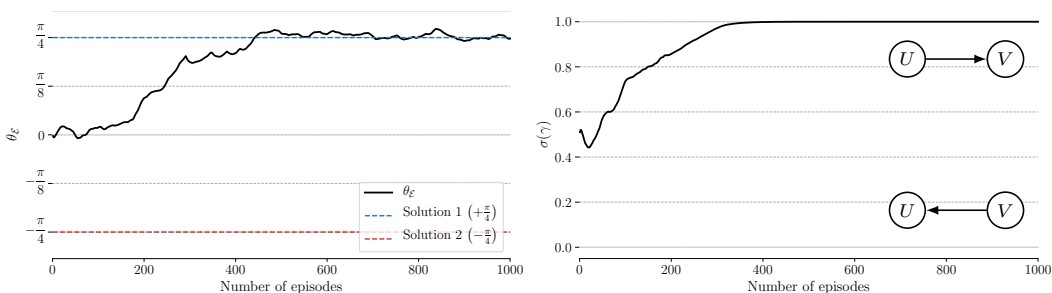

Figure 4: Evolution of structural parameters $\theta_{\mathcal{E}}$ and $\gamma$, as number of episodes increases. Angle of the rotation for the decoder is set to $\theta_{\mathcal{D}} = -\pi/4$, so there are two valid solutions for the angle $\theta_{\mathcal{E}}$ of the encoder: either $\theta_{\mathcal{E}} = \pi/4$, or $\theta_{\mathcal{E}} = -\pi/4$; the model converges to the former solution.

## 5  RELATED WORK

As stated already by Bengio et al. (2013), and clearly demonstrated by Locatello et al. (2019), assumptions, priors, or inductive biases are necessary to identify the underlying explanatory variables. The latter paper (Locatello et al., 2019) also reviews and evaluates recent work on disentangling, and discusses different metrics that have been proposed. Chalupka et al. (2015; 2017) recognize the potential and the challenges underlying causal representation learning. Closely related to our efforts is (Chalupka et al., 2017), which places a strong focus on the coalescence of low (e.g. sensory) level observations (*microvariables*) to higher level causal variables (*macrovariables*), albeit in a more observational setting.

There also exists an extensive literature on learning the structure of Bayesian networks from (observational) data, via score-based methods (Koller & Friedman, 2009). Heckerman et al. (1995); Daly et al. (2011) provide a comprehensive review of these methods. Many of these algorithms are based on greedy-search with local changes to the graphs (Chickering, 2002b), whereas we propose a continuous and fully-differentiable alternative. While most of these approaches only rely on observational data, it is sometimes possible to extend the definition of these scores to interventional data (Hauser & Bühlmann, 2012). The online-likelihood score presented here supports interventional data as its main feature.

Some identifiability results exist for causal models with purely observational data though (Peters et al., 2017), based on specific assumptions on the underlying causal graph. However, causal discovery is more natural under local changes in distributions (Tian & Pearl, 2001), similar to the setting used in this paper. Pearl's seminal work on do-calculus (Pearl, 1995; 2009; Bareinboim & Pearl, 2016) lays the foundation for expressing the impact of interventions on causal graphical models. Here we are proposing a meta-learning objective function for learning the causal structure (without hidden variables), requiring mild assumptions such as localized changes in distributions and faithfulness of the causal graph, in contrast to the stronger assumptions necessary for these identifiability results.

Our work is also related to other recent advances in causation, domain adaptation, and transfer learning. Magliacane et al. (2018) have sought to identify a subset of features that leads to the best predictions for a variable of interest in a source domain, such that the conditional distribution of that variable given these features is the same in the target domain. Zhang et al. (2017) also examine non-stationarity and find that it makes causal discovery easier. Our adaptation procedure, using gradient ascent, is also closely related to gradient-based methods in meta-learning (Finn et al., 2017; Finn, 2018). Alet et al. (2018) proposed a meta-learning algorithm to recover a set of specialized modules, but did not establish any connections to causal mechanisms. More recently, Dasgupta et al. (2019) adopted a meta-learning approach to perform causal inference on purely observational data.

## 6    DISCUSSION & FUTURE WORK

We have established, in very simple bivariate settings, that the rate at which a learner adapts to sparse changes in the distribution of observed data can be exploited to infer the causal structure, and disentangle the causal variables. This relies on the assumption that with the correct causal structure, those distributional changes are localized. We have demonstrated these ideas through some theoretical results, as well as experimental validation. The source code for the experiments is available here: `https://bit.ly/2M6X1al`.

This work is only a first step in the direction of causal structure learning based on the speed of adaptation to modified distributions. On the experimental side, many settings other than those studied here should be considered, with different kinds of parametrizations, richer and larger causal graphs (see already Ke et al. (2019), based on a first version of this paper), or different kinds of optimization procedures. On the theoretical side, much more needs to be done to formally link the locality of interventions to faster adaptation, to clarify the conditions for this to work. Also, more work needs to be done in exploring how the proposed ideas can be used to learn good representations in which the causal variables are disentangled. Scaling up these ideas would permit their application towards improving the way learning agents deal with non-stationarities, and thus improving sample complexity and robustness of these agents.

An extreme view of disentangling is that the explanatory variables should be marginally independent, and many deep generative models (Goodfellow et al., 2016), and Independent Component Analysis models (Hyvärinen et al., 2001; Hyvärinen et al., 2018), are built on this assumption. However, the kinds of high-level variables that we manipulate with natural language are not marginally independent: they are related to each other through statements that are usually expressed in sentences (e.g. a sentence in natural language, or a classical symbolic AI fact or rule), involving only a few concepts at a time. This kind of assumption has been proposed to help discover relevant high-level representations from raw observations, such as the consciousness prior (Bengio, 2017), with the idea that humans focus at any particular time on just a few concepts that are present to our consciousness. The work presented here could provide an interesting meta-learning approach to help learn such encoders outputting causal variables, as well as figure out how the resulting variables are related to each other. In that case, one should distinguish two important assumptions: the first one is that the causal graph is sparse, which a common assumption in structure learning (Schmidt et al., 2007); the second is that the changes in distributions are sparse, which is the focus of this work.

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

## A    RESULTS ON NON-IDENTIFIABILITY OF THE CAUSAL STRUCTURE

Suppose that $A$ and $B$ are two discrete random variables, each taking $N$ possible values. We show here that the maximum likelihood estimation of both models $A \to B$ and $B \to A$ yields the same estimated distribution over $A$ and $B$. The joint likelihood on the training distribution is not sufficient to distinguish the causal model between the two hypotheses. If $p$ is the training distribution, let

$$\theta_i = p(A = i) \qquad\qquad \theta_{j|i} = p(B = j \mid A = i) \qquad (10)$$

$$\eta_j = p(B = j) \qquad\qquad \eta_{i|j} = p(A = i \mid B = j) \qquad (11)$$

Let $\mathcal{D}_{obs}$ be a training dataset. If $N_i^{(A)}$ is the number of samples in $\mathcal{D}_{obs}$ where $A = i$, $N_j^{(B)}$ the number of samples where $B = j$, and $N_{ij}$ the number of samples where $A = i$ and $B = j$, then the maximum likelihood estimator for each parameter is

$$\hat{\theta}_i = N_i^{(A)}/N \qquad\qquad \hat{\theta}_{j|i} = N_{ij}/N_i^{(A)} \qquad (12)$$

$$\hat{\eta}_j = N_j^{(B)}/N \qquad\qquad \hat{\eta}_{i|j} = N_{ij}/N_j^{(B)}. \qquad (13)$$

The estimated distributions for each model $A \to B$ and $B \to A$, under the maximum likelihood estimator, will be equal:

$$\hat{p}(A = i, B = j\,;\, A \to B) = \hat{\theta}_i \hat{\theta}_{j|i} = N_{ij}/N \qquad (14)$$

$$\hat{p}(A = i, B = j\,;\, B \to A) = \hat{\eta}_j \hat{\eta}_{i|j} = N_{ij}/N \qquad (15)$$

To illustrate this result, we also experiment with maximizing the likelihood for each modules for both models $A \to B$ and $B \to A$ with SGD. In Figure A.1, we show the difference in log-likelihoods between these two models, evaluated on training and test data sampled from the same distribution, during training. We can see that while the model $A \to B$ fits the data faster than the other model (corresponding to a positive difference in the figure), both models achieve the same log-likelihoods at convergence. This shows that the two models are indistinguishable, in the limit, based on data sampled from the same distribution, even on test data.

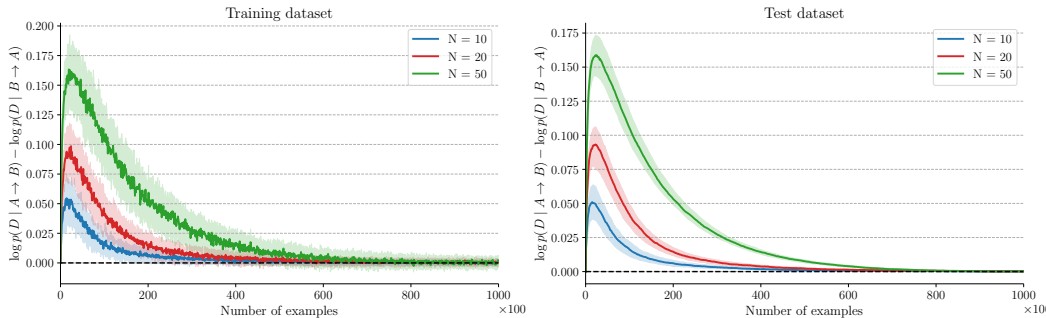

Figure A.1: Difference in log-likelihoods between the two models $A \to B$ and $B \to A$ on training and test data from the same distribution on discrete data, for different values of $N$, the number of discrete values per variable. Once fully trained, both models become indistinguishable from their log-likelihoods only, even on test data. The solid curves represent the median values over 100 different runs, and the shaded areas their 25-75 quantiles.

## B    PROOFS

### B.1    ZERO-GRADIENT UNDER MECHANISM CHANGE

Let us restate Proposition 1 here for convenience:

**Proposition 1.** *Let $G$ be a causal graph, and $p$ a (training) distribution that factorizes according to $G$, with parameters $\theta$. Let $\tilde{p}$ be a second (transfer) distribution that also factorizes according to $G$. If the training and transfer distributions have the same conditional probability distributions for all $V_i$ but a subset $C$ (e.g. the transfer distribution is the result of an intervention on the nodes in $C$):*

$$p(V_i \mid \mathrm{Pa}_G(V_i)) \stackrel{d}{=} \tilde{p}(V_i \mid \mathrm{Pa}_G(V_i)) \qquad \forall V_i \notin C \qquad (16)$$

then the expected gradient w.r.t. the parameters $\theta_i$ such that $V_i \notin C$ of the log-likelihood under the transfer distribution will be zero

$$\forall V_i \notin C, \ \mathbb{E}_{V \sim \tilde{p}}\left[\frac{\partial \log p(V)}{\partial \theta_i}\right] = 0. \tag{17}$$

*Proof.* For $V_i \notin C$, we can simplify the expected gradient as follows:

$$\mathbb{E}_{V \sim \tilde{p}}\left[\frac{\partial \log p(V)}{\partial \theta_i}\right] = \mathbb{E}_{V \sim \tilde{p}}\left[\sum_{j=1}^{n} \frac{\partial}{\partial \theta_i} \log p(V_j \mid \mathrm{Pa}_G(V_j)\,;\,\theta_j)\right] \tag{18}$$

$$= \mathbb{E}_{V \sim \tilde{p}}\left[\frac{\partial}{\partial \theta_i} \log p(V_i \mid \mathrm{Pa}_G(V_i)\,;\,\theta_i)\right] \tag{19}$$

$$= \mathbb{E}_{V \sim \tilde{p}}\left[\frac{\partial}{\partial \theta_i} \log \tilde{p}(V_i \mid \mathrm{Pa}_G(V_i)\,;\,\theta_i)\right] \tag{20}$$

$$= \mathbb{E}_{V \sim \tilde{p}}\left[\sum_{j=1}^{n} \frac{\partial}{\partial \theta_i} \log \tilde{p}(V_j \mid \mathrm{Pa}_G(V_j)\,;\,\theta_j)\right] \tag{21}$$

$$= \mathbb{E}_{V \sim \tilde{p}}\left[\frac{\partial \log \tilde{p}(V)}{\partial \theta_i}\right] = 0 \tag{22}$$

where Equation (20) arises from our assumption that the conditional distribution of $V_i$ given its parents in $G$ does not change between the training distribution $p$ and the transfer distribution $\tilde{p}$. Moreover, the last equality arises from the marginalization

$$\sum_{\mathbf{v}} \tilde{p}(\mathbf{v}) = 1 \tag{23}$$

$\blacksquare$

### B.2 Gradient of the structural parameter

Let us restate Proposition 2 here for convenience:

**Proposition 2.** *The gradient of the negative log-likelihood of the transfer data $\mathcal{D}_{int}$ in Equation (5) wrt. the structural parameter $\gamma$ is given by*

$$\frac{\partial \mathcal{R}}{\partial \gamma} = p(A \to B) - p(A \to B \mid \mathcal{D}_{int}), \tag{24}$$

*where $p(A \to B \mid \mathcal{D}_{int})$ is the posterior probability of the hypothesis $A \to B$ (when the alternative is $B \to A$). Furthermore, this can be equivalently written as*

$$\frac{\partial \mathcal{R}}{\partial \gamma} = \sigma(\gamma) - \sigma(\gamma + \Delta), \tag{25}$$

*where $\Delta = \log \mathcal{L}_{A \to B}(\mathcal{D}_{int}) - \log \mathcal{L}_{B \to A}(\mathcal{D}_{int})$ is the difference between the online log-likelihoods of the two hypotheses on the transfer data $\mathcal{D}_{int}$.*

*Proof.* First note that, using Bayes rule,

$$p(A \to B \mid \mathcal{D}_{int}) = \frac{p(\mathcal{D}_{int} \mid A \to B)p(A \to B)}{p(\mathcal{D}_{int} \mid A \to B)p(A \to B) + p(\mathcal{D}_{int} \mid B \to A)p(B \to A)} \tag{26}$$

$$= \frac{\mathcal{L}_{A \to B}(\mathcal{D}_{int})\sigma(\gamma)}{\mathcal{L}_{A \to B}(\mathcal{D}_{int})\sigma(\gamma) + \mathcal{L}_{B \to A}(\mathcal{D}_{int})(1 - \sigma(\gamma))} \tag{27}$$

$$= \frac{\mathcal{L}_{A \to B}(\mathcal{D}_{int})\sigma(\gamma)}{M} \tag{28}$$

where $M = \mathcal{L}_{A \to B}(\mathcal{D}_{int}) \sigma(\gamma) + \mathcal{L}_{B \to A}(\mathcal{D}_{int})(1 - \sigma(\gamma))$ is the online likelihood of the transfer data under the mixture, so that the regret is $\mathcal{R}(\mathcal{D}_{int}) = -\log M$. For Equation (27), note that if $\mathcal{D}_{int} = \{a_t, b_t\}_{t=1}^{T}$,

$$p(\mathcal{D}_{int} \mid A \to B) = \prod_{t=1}^{T} p(a_t, b_t \mid A \to B, \{a_s, b_s\}_{s=1}^{t-1}) \tag{29}$$

$$= \prod_{t=1}^{T} p(a_t, b_t \mid A \to B \,;\, \theta_{A \to B}^{(t)}) = \mathcal{L}_{A \to B}(\mathcal{D}_{int}) \tag{30}$$

where $\theta_{A \to B}^{(t)}$ encapsulates the information about the previous datapoints $\{a_s, b_s\}_{s=1}^{t-1}$ in the graph $A \to B$, through some adaptation procedure. Since we only consider the two hypotheses $A \to B$ and $B \to A$, we also have

$$p(B \to A \mid \mathcal{D}_{int}) = 1 - p(A \to B \mid \mathcal{D}_{int}) = \frac{\mathcal{L}_{B \to A}(\mathcal{D}_{int})(1 - \sigma(\gamma))}{M} \tag{31}$$

Therefore, the gradient of the regret wrt. the structural parameter $\gamma$ is

$$\frac{\partial \mathcal{R}}{\partial \gamma} = -\frac{1}{M} \left[ \sigma(\gamma)(1 - \sigma(\gamma))\mathcal{L}_{A \to B}(\mathcal{D}_{int}) - \sigma(\gamma)(1 - \sigma(\gamma))\mathcal{L}_{B \to A}(\mathcal{D}_{int}) \right] \tag{32}$$

$$= \sigma(\gamma)p(B \to A \mid \mathcal{D}_{int}) - (1 - \sigma(\gamma))p(A \to B \mid \mathcal{D}_{int}) \tag{33}$$

$$= \sigma(\gamma) - \sigma(\gamma)p(B \to A \mid \mathcal{D}_{int}) - p(A \to B \mid \mathcal{D}_{int}) + \sigma(\gamma)p(A \to B \mid \mathcal{D}_{int}) \tag{34}$$

$$= \sigma(\gamma) - p(A \to B \mid \mathcal{D}_{int}) \tag{35}$$

$$= p(A \to B) - p(A \to B \mid \mathcal{D}_{int}) \tag{36}$$

which concludes the first part of the proof. Moreover, given Equation (35), it is sufficient to show that $p(A \to B \mid \mathcal{D}_{int}) = \sigma(\gamma + \Delta)$ to prove the equivalent formulation in Equation (25). Using the logit function $\sigma^{-1}(z) = \log \frac{z}{1-z}$, and the expression in Equation (28), we have

$$\sigma^{-1}(p(A \to B \mid \mathcal{D}_{int})) = \log \frac{\sigma(\gamma)\mathcal{L}_{A \to B}(\mathcal{D}_{int})}{M - \sigma(\gamma)\mathcal{L}_{A \to B}(\mathcal{D}_{int})} \tag{37}$$

$$= \log \frac{\sigma(\gamma)\mathcal{L}_{A \to B}(\mathcal{D}_{int})}{(1 - \sigma(\gamma))\mathcal{L}_{B \to A}(\mathcal{D}_{int})} \tag{38}$$

$$= \underbrace{\log \frac{\sigma(\gamma)}{1 - \sigma(\gamma)}}_{= \gamma} + \underbrace{\log \mathcal{L}_{A \to B}(\mathcal{D}_{int}) - \log \mathcal{L}_{B \to A}(\mathcal{D}_{int})}_{= \Delta} \tag{39}$$

$$= \gamma + \Delta. \tag{40}$$

$\blacksquare$

### B.3 Convergence point of Gradient Descent on the structural parameter

Let us restate Proposition 3 here for convenience:

**Proposition 3.** *With stochastic gradient descent (and an appropriately decreasing learning rate) on $\mathbb{E}_{\mathcal{D}_{int}}[\mathcal{R}(\mathcal{D}_{int})]$, where the gradient steps are given by Proposition 2, the structural parameter converges towards*

$$\begin{aligned} \sigma(\gamma) \to 1 \quad &\text{if } \mathbb{E}_{\mathcal{D}_{int}}[\mathcal{L}_{A \to B}(\mathcal{D}_{int})] > \mathbb{E}_{\mathcal{D}_{int}}[\mathcal{L}_{B \to A}(\mathcal{D}_{int})] \\ \text{or } \sigma(\gamma) \to 0 \quad &\text{otherwise} \end{aligned} \tag{41}$$

*Proof.* We are going to consider the fixed point of gradient descent (a point where the gradient is zero), since we already know that SGD converges with an appropriately decreasing learning rate. Let us introduce some notations to simplify the algebra: let $p = \sigma(\gamma)$, $M = p\mathcal{L}_{A \to B}(\mathcal{D}_{int}) + (1 - p)\mathcal{L}_{B \to A}(\mathcal{D}_{int})$, so that the regret is $\mathcal{R}(\mathcal{D}_{int}) = -\log M$. We define $P_1$ and $P_2$ as (see also the proof in Appendix B.2)

$$P_1 = \frac{p\mathcal{L}_{A \to B}(\mathcal{D}_{int})}{M} = p(A \to B \mid \mathcal{D}_{int}) \qquad P_2 = \frac{(1 - p)\mathcal{L}_{B \to A}(\mathcal{D}_{int})}{M} = 1 - P_1 \tag{42}$$

Framing the stationary point in terms of $p$ rather than $\gamma$ gives us a constrained optimization problem, with inequality constraints $-p \leq 0$ and $p - 1 \leq 0$, and no equality constraint.

$$\min_p \ \mathbb{E}_{\mathcal{D}_{int}}[\mathcal{R}(\mathcal{D}_{int})] \tag{43}$$

$$\text{s.t.} \ -p \leq 0 \tag{44}$$

$$p - 1 \leq 0 \tag{45}$$

Applying the KKT conditions to this problem, with constraint functions $-p$ and $p - 1$, gives us

$$\mathbb{E}_{\mathcal{D}_{int}}\left[\frac{\partial \mathcal{R}}{\partial p}\right] = -\mu_1 + \mu_2 \tag{46}$$

$$\mu_i \geq 0 \qquad \text{for } i = 1, 2 \tag{47}$$

$$\mu_1 p = 0 \tag{48}$$

$$\mu_2(p - 1) = 0 \tag{49}$$

We already see from equations (48) & (49) that if $p \in (0, 1)$ (i.e. excluding 0 and 1), we must have $\mu_1 = \mu_2 = 0$, that is

$$\mathbb{E}_{\mathcal{D}_{int}}\left[\frac{\partial \mathcal{R}}{\partial p}\right] = 0. \tag{50}$$

Let us study that case first, and show that it leads to an inconsistent set of equations (thus, forcing the solution to be either $p = 0$ or $p = 1$). Let us rewrite the gradient to highlight $p$ in it (using Proposition 2):

$$\frac{\partial \mathcal{R}}{\partial p} = \frac{1}{p(1-p)} \left(p - p(A \to B \mid \mathcal{D}_{int})\right) \tag{51}$$

$$= \frac{1}{p(1-p)} \left[p - \frac{p\mathcal{L}_{A \to B}(\mathcal{D}_{int})}{M}\right] \tag{52}$$

$$= \frac{1}{p(1-p)} \frac{p(p\mathcal{L}_{A \to B}(\mathcal{D}_{int}) + (1-p)\mathcal{L}_{B \to A}(\mathcal{D}_{int})) - p\mathcal{L}_{A \to B}(\mathcal{D}_{int})}{M} \tag{53}$$

$$= \frac{\mathcal{L}_{B \to A}(\mathcal{D}_{int}) - \mathcal{L}_{A \to B}(\mathcal{D}_{int})}{M} \tag{54}$$

This derivation is valid since we assume that $p \in (0, 1)$. Suppose that $p \neq 0$; multiplying both sides of Equation (50) by $p$ gives

$$0 = \mathbb{E}_{\mathcal{D}_{int}}\left[\frac{p(\mathcal{L}_{B \to A}(\mathcal{D}_{int}) - \mathcal{L}_{A \to B}(\mathcal{D}_{int}))}{M}\right] \tag{55}$$

$$= \mathbb{E}_{\mathcal{D}_{int}}\left[\frac{p\mathcal{L}_{B \to A}(\mathcal{D}_{int})}{M} - P_1\right] \tag{56}$$

$$= \mathbb{E}_{\mathcal{D}_{int}}\left[\frac{\mathcal{L}_{B \to A}(\mathcal{D}_{int})}{M} - P_2 - P_1\right] \tag{57}$$

$$= \mathbb{E}_{\mathcal{D}_{int}}\left[\frac{\mathcal{L}_{B \to A}(\mathcal{D}_{int})}{M} - 1\right] \tag{58}$$

For this equation to be satisfied, we need $\mathcal{L}_{B \to A} = M$ almost surely, since $\mathcal{L}_{B \to A}(\mathcal{D}_{int}) \leq M$ by construction. This would, however, correspond to $p = 0$, which contradicts our assumption. Similarly, assuming that $p \neq 1$, we can also multiply both sides of Equation (50) by $1 - p$ and get

$$0 = \mathbb{E}_{\mathcal{D}_{int}}\left[\frac{(1-p)(\mathcal{L}_{B \to A}(\mathcal{D}_{int}) - \mathcal{L}_{A \to B}(\mathcal{D}_{int}))}{M}\right] \tag{59}$$

$$= \mathbb{E}_{\mathcal{D}_{int}}\left[P_2 - \frac{(1-p)\mathcal{L}_{A \to B}(\mathcal{D}_{int})}{M}\right] \tag{60}$$

$$= \mathbb{E}_{\mathcal{D}_{int}}\left[P_2 + P_1 - \frac{\mathcal{L}_{A \to B}(\mathcal{D}_{int})}{M}\right] \tag{61}$$

$$= \mathbb{E}_{\mathcal{D}_{int}}\left[1 - \frac{\mathcal{L}_{A \to B}(\mathcal{D}_{int})}{M}\right] \tag{62}$$

Again, this can only be true if $\mathcal{L}_{A \to B} = M$ almost surely, meaning that $p = 1$, contradicting our assumption. We conclude that the solutions $p \in (0, 1)$ are not possible because they would lead to inconsistent conclusions, which leaves only $p = 0$ or $p = 1$. ∎

## C  RESULTS ON LEARNING WHICH IS CAUSE AND WHICH IS EFFECT

In order to assess the performance of our meta-learning algorithm, we applied it on generated data from three different domains: discrete random variables, multimodal continuous random variables and multivariate Gaussian-distributed variables. In this section, we describe the setups for all three experiments, along with additional results to complement the results descrbed in Section 3.3. Note that in all these experiments, we fix the ground-truth structure as $A \to B$, and only perform interventions on the cause $A$.

### C.1  DISCRETE VARIABLES WITH TABULAR REPRESENTATION

We consider a bivariate model, where both random variables are sampled from a categorical distribution. The underlying ground-truth model can be described as

$$A \sim \text{Categorical}(\pi_A) \tag{63}$$
$$B \mid A = a \sim \text{Categorical}(\pi_{B|a}), \tag{64}$$

with $\pi_A$ a probability vector of size $N$, and $\pi_{B|a}$ a probability vector of size $N$, which depends on the value of the variable $A$. In our experiment, each random variable can take one of $N = 10$ or $N = 100$ values. Since we are working with only two variables, the only two possible models are:

- *Model $A \to B$*: $p(A, B) = p(A)p(B \mid A)$
- *Model $B \to A$*: $p(A, B) = p(B)p(A \mid B)$

We build 4 different modules, corresponding to every possible marginal and conditional distributions. Here, we use multinomial logistic Conditional Probability Distributions (Koller & Friedman, 2009). The modules' definition, and their corresponding parameters, are shown in Table C.1.

Table C.1: Description of the 2 models, with the parametrization of each module, for a bivariate model with discrete random variables. *Model $A \to B$* and *Model $B \to A$* both have the same number of parameters $N^2 + N$.

| Distribution | | Module | Parameters | Dimension |
|---|---|---|---|---|
| *Model* | $p(A)$ | $p(x_A = i\,;\theta_A) = [\text{softmax}(\theta_A)]_i$ | $\theta_A$ | $N$ |
| $A \to B$ | $p(B \mid A)$ | $p(x_B = j \mid x_A = i\,;\theta_{B|A}) = [\text{softmax}(\theta_{B|A}(i))]_j$ | $\theta_{B|A}$ | $N^2$ |
| *Model* | $p(B)$ | $p(x_B = j\,;\theta_B) = [\text{softmax}(\theta_B)]_j$ | $\theta_B$ | $N$ |
| $B \to A$ | $p(A \mid B)$ | $p(x_A = i \mid x_B = j\,;\theta_{A|B}) = [\text{softmax}(\theta_{A|B}(j))]_i$ | $\theta_{A|B}$ | $N^2$ |

In order to get a set of initial parameters, we first train all 4 modules on a training distribution ($p$ in the main text). This distribution corresponds to a fixed choice of $\pi_A^{(1)}$ and $\pi_{B|a}$ (for all N possible values of $a$). The superscript in $\pi_A^{(1)}$ emphasizes the fact that this defines the distribution prior to an intervention, with the mechanism $p(B \mid A)$ being unchanged by the intervention. These probability vectors are sampled randomly from a uniform Dirichlet distribution:

$$\pi_A^{(1)} \sim \text{Dirichlet}(\mathbf{1}_N) \tag{65}$$
$$\pi_{B|a} \sim \text{Dirichlet}(\mathbf{1}_N) \qquad \forall a \in [1, N]. \tag{66}$$

Given this training distribution, we can sample a large dataset of samples $\mathcal{D}_{obs} = \{a_i, b_i\}_{i=1}^m$ for the ground truth model, using ancestral sampling. Using $\mathcal{D}_{obs}$, we can train all 4 modules using gradient ascent on the log-likelihood (or any other advanced first-order optimizer, like RMSprop). The parameters $\theta_A$, $\theta_{B|A}$, $\theta_B$ & $\theta_{A|B}$ of the maximum likelihood estimate will be used as the initial parameters for the adaptation on the new transfer distribution.

Similar to the way we defined the training distribution, we can define a transfer distribution ($\tilde{p}$ in the main text) as an intervention on the random variable $A$. In this experiment, this accounts for changing the distribution of $A$, that is with a new probability vector $\pi_A^{(2)}$, also sampled from a uniform Dirichlet distribution

$$\pi_A^{(2)} \sim \text{Dirichlet}(\mathbf{1}_N). \tag{67}$$

To perform adaptation on the transfer distribution, we also sample a smaller transfer dataset $\mathcal{D}_{int} = \{a_t, b_t\}_{t=1}^T$, with $T \ll m$. In our experiment, we used $T = 20$ datapoints, following the observation from Section 2.1.

## C.2 DISCRETE VARIABLES WITH MLP PARAMETRIZATION

We consider a bivariate model, similar to the one defined in Appendix C.1, where each random variable is sampled from a categorical distribution. Instead of expressing the CPDs in tabular form, we use structured CPDs, parametrized with multi-layer perceptrons (MLPs). In our experiment, all the MLPs have only one hidden layer with $H = 8$ hidden units, with a ReLU non-linearity, and the output layer has a softmax non-linearity. To avoid any modeling bias, we assume that the ground-truth model is also parametrized by MLPs, such that

$$A \sim \text{Categorical}(\text{MLP}(\mathbf{0}\,;\,W_A)) \tag{68}$$
$$B \mid A = a \sim \text{Categorical}(\text{MLP}(\mathbf{1}[a]\,;\,W_B)) \tag{69}$$

where $\mathbf{0}$ is a vector of size $N$ will all zeros, and $\mathbf{1}[a]$ is a one-hot vector of size $N$. $W_A$ and $W_B$ summarize the parameters of the ground truth model, with the weights and biases for the 2 layers. Similar to the tabular representation, we define 4 different modules, this time using MLPs. Their definition, as well as their corresponding parameters, are shown in Table C.2.

Table C.2: Description of the 2 models, with the parametrization of each module, for a bivariate model with discrete random variables, and MLP parametrization. *Model $A \to B$* and *Model $B \to A$* both have the same number of parameters $3NH + 2(N + H)$.

| Distribution | | Module | Parameters | Dimension |
|---|---|---|---|---|
| *Model* | $p(A)$ | $p(x_A = i\,;\,\theta_A) = [\text{MLP}(\mathbf{0}\,;\,\theta_A)]_i$ | $\theta_A$ | $NH + H + N$ |
| $A \to B$ | $p(B \mid A)$ | $p(x_B = j \mid x_A\,;\,\theta_{B\mid A}) = [\text{MLP}(\mathbf{1}[x_A]\,;\,\theta_{B\mid A})]_j$ | $\theta_{B\mid A}$ | $2NH + H + N$ |
| *Model* | $p(B)$ | $p(x_B = j\,;\,\theta_B) = [\text{MLP}(\mathbf{0}\,;\,\theta_B)]_j$ | $\theta_B$ | $NH + H + N$ |
| $B \to A$ | $p(A \mid B)$ | $p(x_A = i \mid x_B\,;\,\theta_{A\mid B}) = [\text{MLP}(\mathbf{1}[x_B]\,;\,\theta_{A\mid B})]_i$ | $\theta_{A\mid B}$ | $2NH + H + N$ |

Again, to define the training distribution, we first fix the parameters $W_A^{(1)}$ and $W_B$. We use randomly initialized networks for the training distribution, with the parameters sampled using the He initialization. We train all the modules using maximum likelihood on a large dataset of training samples $\mathcal{D}_{obs}$, to get the initial set of parameters for the adaptation on the transfer distribution.

We also define a transfer distribution as the result of an intervention on $A$. In this experiment, this means sampling a new set of parameters $W_A^{(2)}$, still as a randomly initialized network. We sample a transfer dataset $\mathcal{D}_{int} = \{a_t, b_t\}_{t=1}^T$, with $T = 20$ datapoints.

## C.3 CONTINUOUS MULTIMODAL VARIABLES

Consider a family of joint distributions $p_\mu(A, B)$ over the causal variables $A$ and $B$, defined by the following structural causal model (SCM):

$$A \sim p_\mu(A) = \mathcal{N}(\mu, \sigma^2 = 4) \tag{70}$$
$$B := f(A) + N_B \qquad\qquad N_B \sim \mathcal{N}(0, 1), \tag{71}$$

where $f$ is a randomly generated spline, and the noise $N_B$ is sampled iid. from the unit Gaussian distribution. To obtain the spline, we sample $K$ points $\{x_k\}_{k=1}^K$ uniformly spaced from the interval $[-8, 8]$, and another $K$ points $\{y_k\}_{k=1}^K$ uniformly randomly from the interval $[-8, 8]$. This yields $K$

pairs $\{x_k, y_k\}_{k=1}^K$, which make the knots of a second-order spline. We choose $K = 8$ points in our experiments.

The conditional distributions $p(B \mid A)$ and $p(A \mid B)$ are parametrized as 2-layer Mixture Density Networks (MDNs; Bishop, 1994), with 32 hidden units and 10 components. The marginal distributions $p(A)$ and $p(B)$ are parametrized as Gaussian Mixture Models (GMMs), also with 10 components. The definition of the different modules, as well as their corresponding parameters, are shown in Table C.3.

Table C.3: Description of the 2 models, with the parametrization of each module, for a bivariate model with continuous multimodal variables. *Model $A \to B$* and *Model $B \to A$* both have the same number of parameters 2,140.

| Distribution | | Module | Parameters | Dimension |
|---|---|---|---|---|
| *Model* | $p(A)$ | $p(x_A \,; \theta_A) = \mathrm{GMM}(x_A \,; \theta_A)$ | $\theta_A$ | 30 |
| $A \to B$ | $p(B \mid A)$ | $p(x_B \mid x_A \,; \theta_{B\mid A}) = \mathrm{MDN}(x_B, x_A \,; \theta_{B\mid A})$ | $\theta_{B\mid A}$ | 2,110 |
| *Model* | $p(B)$ | $p(x_B \,; \theta_B) = \mathrm{GMM}(x_B \,; \theta_B)$ | $\theta_B$ | 30 |
| $B \to A$ | $p(A \mid B)$ | $p(x_A \mid x_B \,; \theta_{A\mid B}) = \mathrm{MDN}(x_A, x_B \,; \theta_{A\mid B})$ | $\theta_{A\mid B}$ | 2,110 |

We select $p_0(A, B)$ as the training distribution, from which we sample a large dataset $\mathcal{D}_{obs}$ using ancestral sampling. Similar to the earlier experiments, this dataset is used to get the initial set of parameters for the adaptation on the transfer distribution. The MDNs are fitted with gradient descent, while the GMMs are learned via Expectation Maximization. The transfer distribution is the result of an intervention on $A$, where we shift the distribution $p_\mu(A)$ with $\mu$ sampled uniformly in $[-1, 1]$. In Figure C.1, we plot samples from the training distribution ($\mu = 0$), as well as two transfer distributions ($\mu = \pm 4$).

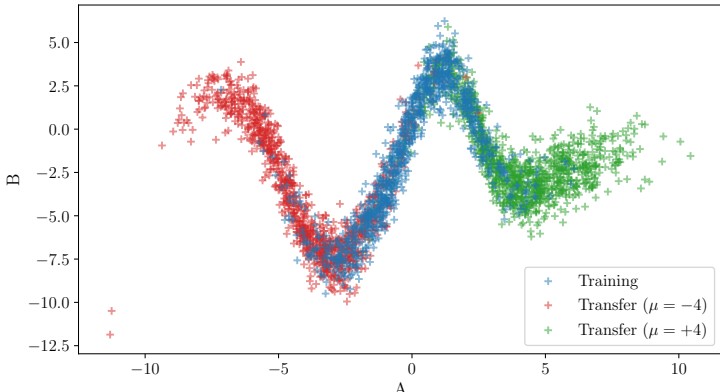

Figure C.1: Samples from the training (blue) and transfer (red and green) distributions, from an SCM generated with the procedure described above. The red datapoints are sampled from $p_{-4}(A, B)$, the green datapoints from $p_4(A, B)$, and the blue datapoints from $p_0(A, B)$.

The structural regret $\mathcal{R}(\gamma)$ is now minimized with respect to $\gamma$ for 500 iterations (updates of $\gamma$). In the notation of Algorithm 1, these are the iterations over the number of episodes. Figure C.2 shows the evolution of $\sigma(\gamma)$ as training progresses. This is expected, given that we expect the causal model to perform better on the transfer distributions, i.e. we expect $\mathcal{L}_{A\to B} > \mathcal{L}_{B\to A}$ in expectation. Consequently, assigning a larger weight to $\mathcal{L}_{A\to B}$ optimizes the objective (see Proposition 3).

Finally as a sanity check, we test the experimental set-up described above on a linear SCM with additive Gaussian noise. In this setting, it is well known that the causal structure cannot be discovered from observations alone Peters et al. (2017) and one must rely on the transfer distribution tell cause from effect.

To that end, we repeat the experiment in Figure C.2 with the following amendments: (a) we replace the non-linear spline with a linear curve (Figure C.3), and (b) in addition to training the structural

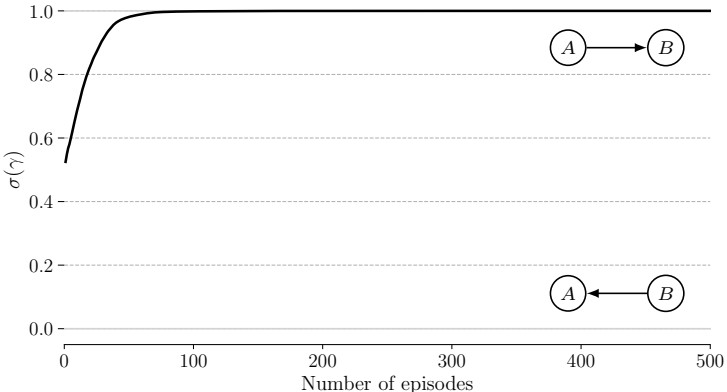

Figure C.2: Evolution of the sigmoid of the structural parameter $\sigma(\gamma)$, with the number of episodes (meta-training iterations). The belief of $A \to B$ being the correct causal model increases as the number of episodes increases.

parameter by adapting the $A \to B$ and $B \to A$ models to multiple interventional distributions, we train it by "adapting" the said model to the train distribution, where the latter serves as a baseline.

Figure C.4 shows that using multiple transfer (i.e. interventional) distributions ("With Interventions") enables causal discovery, as opposed to the model trained with a single observational distribution. This confirms that our method indeed relies on the interventional distributions to discover the causal structure.

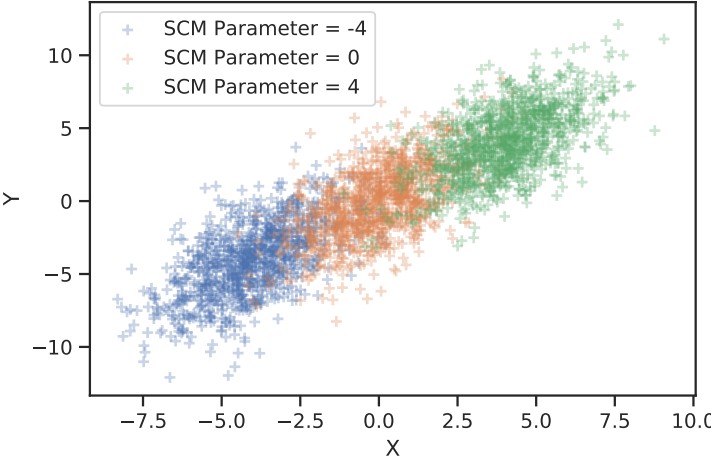

Figure C.3: Samples from a linear SCM, showing training (orange) and two transfer distributions (blue and green).

## C.4 LINEAR GAUSSIAN MODEL

In this experiment, the two variables $A$ and $B$ are vector-valued, taking values in $\mathbb{R}^d$. The ground-truth causal model is given by

$$A \sim \mathcal{N}(\mu, \Sigma) \tag{72}$$

$$B := \beta_1 A + \beta_0 + N_B \qquad\qquad N_B \sim \mathcal{N}(0, \widetilde{\Sigma}), \tag{73}$$

where $\mu \in \mathbb{R}^d$, $\beta_0 \in \mathbb{R}^d$ and $\beta_1 \in \mathbb{R}^{d \times d}$. $\Sigma$ and $\widetilde{\Sigma}$ are two $d \times d$ covariance matrices. In our experiment, $d = 100$. Once again, we want to identify the correct causal direction between $A$ and $B$.

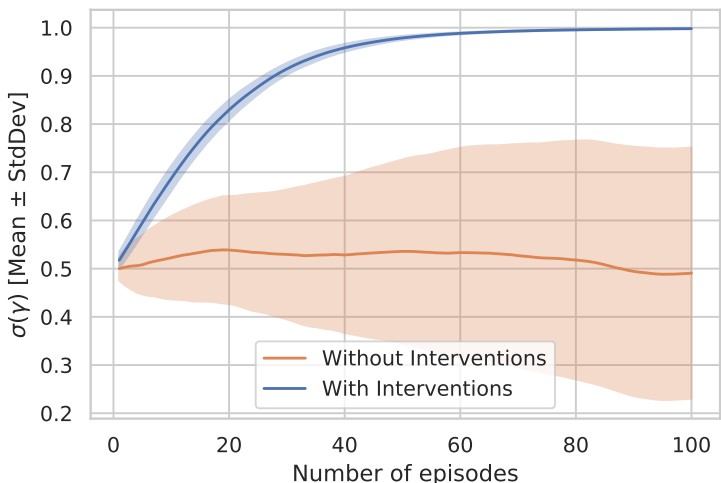

Figure C.4: Evolution of the sigmoid of the structural parameter $\sigma(\gamma)$, with the number of episodes (meta-training iterations) in case of a linear model with additive Gaussian noise. The blue curve corresponds to the setting where we make use of interventions, whereas the orange curve corresponds to one where we do not (i.e. use a single distribution). The shaded bands show the standard deviation over 40 runs (of both pre- and meta-training). We find (as expected) that causal discovery fails without interventions but succeeds when transfer distributions are available.

To do so, we consider two models $A \to B$ and $B \to A$ parametrized with Gaussian distributions. The details of the modules' definitions, as well as their parameters, is given in Table C.4. Note that each covariance matrix is parametrized using the Cholesky decomposition.

Table C.4: Description of the 2 models, with the parametrization of each module, for a bivariate model with linear Gaussian variables. *Model $A \to B$* and *Model $B \to A$* both have the same number of parameters $2d^2 + 3d$.

| Distribution | | Module | Parameters | Dimension |
|---|---|---|---|---|
| *Model* | $p(A)$ | $p(x_A \,; \theta_A) = \mathcal{N}(x_A \mid \mu_A, \Sigma_A)$ | $\mu_A, \Sigma_A$ | $d(d+1)/2 + d$ |
| $A \to B$ | $p(B \mid A)$ | $p(x_B \mid x_A \,; \theta_{B\mid A}) = \mathcal{N}(x_B \mid W_1 x_A + W_0, \Sigma_{B\mid A})$ | $W_1, W_0, \Sigma_{B\mid A}$ | $3d(d+1)/2$ |
| *Model* | $p(B)$ | $p(x_B \,; \theta_B) = \mathcal{N}(x_B \mid \mu_B, \Sigma_B)$ | $\mu_B, \Sigma_B$ | $d(d+1)/2 + d$ |
| $B \to A$ | $p(A \mid B)$ | $p(x_A \mid x_B \,; \theta_{A\mid B}) = \mathcal{N}(x_A \mid V_1 x_B + V_0, \Sigma_{A\mid B})$ | $V_1, V_0, \Sigma_{A\mid B}$ | $3d(d+1)/2$ |

To build the training distribution, we draw $\mu^{(1)}$, $\beta_0$ and $\beta_1$ from a Gaussian distribution, and $\Sigma^{(1)}$ and $\widetilde{\Sigma}$ from an inverse Wishart distribution. The transfer distribution is the result of an intervention on $A$, meaning that the marginal $\tilde{p}(A)$ changes. To do so, we sample new parameters $\mu^{(2)}$ from a Gaussian distribution, and $\Sigma^{(2)}$ from an inverse Wishart distribution as well.

Unlike the previous experiments, we are not conduction any pre-training on actual data from the training distribution. Instead, we fix the parameters of both models to their exact values, according to the ground truth distribution. For Model $A \to B$, this can be done easily. For the Model $B \to A$, we compute the exact parameters analytically using Bayes rule. This can be seen as the maximum likelihood estimate in the limit of infinite data. In Figure C.5, we show that, after 200 episodes, $\sigma(\gamma)$ converges to 1, indicating the success of the method on this particular task.

## C.5 EXPERIMENTS WITH SOFT INTERVENTION

In this section, we describe an experimental setting where the conditional $p(B \mid A)$ is perturbed while the distribution of the cause, $p(A)$, is left unchanged. To that end, consider a set-up similar to that in Section C.3:

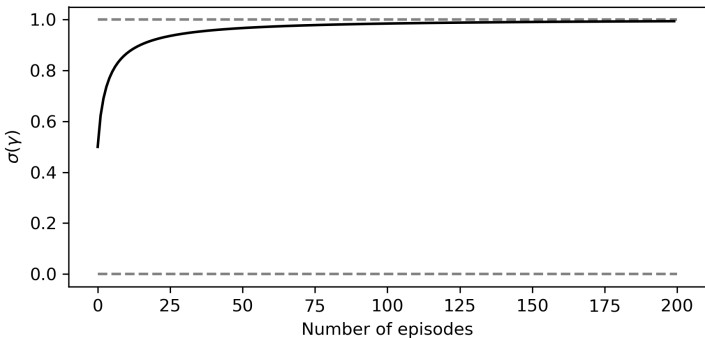

Figure C.5: Convergence of the causal belief (to the correct answer) as a function of the number of meta-learning episodes, for the linear Gaussian experiments.

$$A \sim p_\mu(A) = \mathcal{U}(-8, 8) \tag{74}$$
$$B := f_0(A) + N_B \qquad\qquad N_B \sim \mathcal{N}(0, 1), \tag{75}$$

where $f_0$ is a randomly generated spline and $N_B$ is sampled iid. from the unit Gaussian distribution and the cause variable $A$ is sampled from the uniform distribution supported on $[-8, 8]$.

To induce soft-interventions, we modify the SCM as follows. Consider the knots $\{a_i, b_i\}_{i=1}^5$ of the order 3 spline $f_0$; we obtain a new spline $f_{int}$ by randomly perturbing the $b$-coordinate of the knots, where the perturbations are sampled from another uniform distribution[1]. Using the perturbed spline $f_{int}$ instead of $f_0$ in Equation (74) results in a new SCM, from which we generate a single transfer distribution (i.e. for a single episode). In Figure C.6 we plot samples from three such transfer distributions.

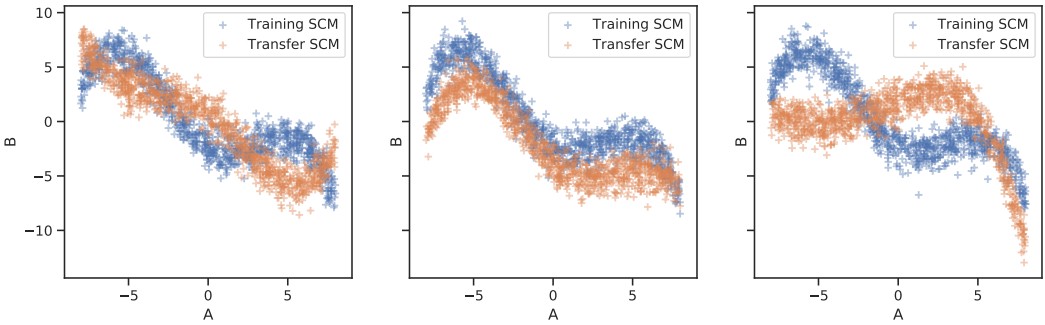

Figure C.6: Samples from different training (blue) and transfer (orange) distributions, from SCMs generated with the procedure described above, namely: all transfer SCMs (orange) are obtained by soft-intervening of the underlying training SCM (blue).

The models used are identical to those detailed in Appendix C.3 and are trained on the training SCM (corresponding base-spline $f_0$) with a large amount of samples ($\approx$ 3,000k). The meta-training procedure differs in that (a) in every transfer episode, we create a new spline $f_{int}$ and sample a transfer distribution $\mathcal{D}_{int}$ from the corresponding SCM, and (b) we use the following measure of adaptation:

$$\mathcal{L}_G(\mathcal{D}_{int}) = \exp\left[\log p(\mathcal{D}_{int} \mid \theta_G^{(T)}) - \log p(\mathcal{D}_{int} \mid \theta_G^{(0)})\right] \tag{76}$$

where $G$ is one of $A \to B$ or $B \to A$. The meta-transfer objective in Equation (5) remains the same.

Figure C.7 shows the evolution of the $\sigma(\gamma)$ as the training progresses, and we find that the structural parameter correctly converges to 1, representing the correct causal graph $A \to B$.

---

[1]The scale of the perturbation is 0.5 times that of the original knots.

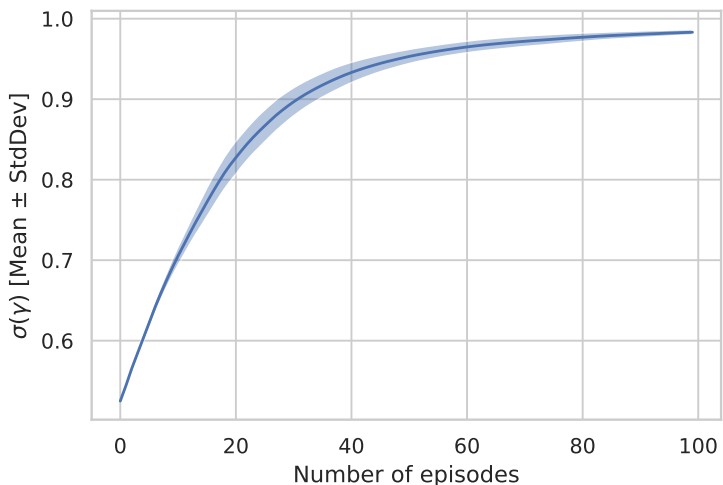

Figure C.7: Convergence of the causal belief (to the correct answer) as a function of the number of meta-learning episodes, for the the experiments with soft interventions. The error band is over 5 different runs.

**Failure case**  In addition to the result above, we also observed that using soft interventions on the effect $B$ instead on changes on the marginal $p(A)$ was sometimes failing to recover the correct causal graph. Instead, the anti-causal graph (here $B \rightarrow A$) was found, with high confidence. We describe here one such experiment where using the meta-transfer objective failed at recovering the correct causal graph.

Our experimental setting is similar to the one described in Appendix C.1. However instead of changing the marginal $p(A)$, the conditional distribution $p(B \mid A)$ changes and $p(A)$ remains unchanged. Following the notations in Appendix C.1, we have

$$\pi_A \sim \text{Dirichlet}(\mathbf{1}_N) \tag{77}$$

$$\pi_{B|a}^{(1)} \sim \text{Dirichlet}(\mathbf{1}_N) \quad \& \quad \pi_{B|a}^{(2)} \sim \text{Dirichlet}(\mathbf{1}_N) \qquad \forall a \in [1, N], \tag{78}$$

where $\pi_{B|a}^{(1)}$ are the parameters of the conditional distribution before intervention, and $\pi_{B|a}^{(2)}$ its parameters after intervention. We again sample data from both the training and transfer distributions to get datasets $\mathcal{D}_{obs}$ and $\mathcal{D}_{int}$. The different modules and their corresponding parameters are defined in Table C.1.

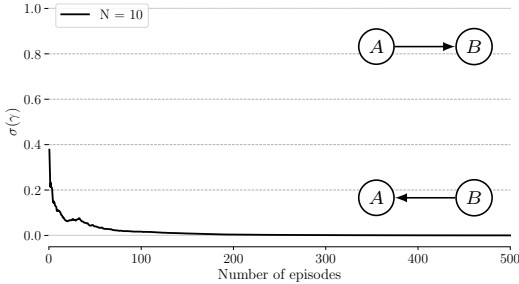

Figure C.8: Evolution of the belief that $A \rightarrow B$ is the correct causal model, as the number of episodes increases, starting with an equal belief for both hypotheses, under soft interventions on the effect $B$.

In Figure C.8, we show the evolution of the structural parameter $\sigma(\gamma)$, the model's belief that $A \rightarrow B$ is the correct causal model, as the number of episodes increases. Unlike our previous experiments in Section 3.3, the structural parameter now converges to $\sigma(\gamma) \rightarrow 0$, corresponding to a strong belief that the model is $B \rightarrow A$. We are therefore unable to recover the correct causal graph here under the

assumption that $p(B \mid A)$ changes. Note that here the parameter counting argument from Section 2.2 clearly does not hold anymore, since the modules all use a tabular representation, and both models require the same order $O(N^2)$ of updates to adapt to a transfer distribution.

## D  RESULTS ON REPRESENTATION LEARNING

The true latent causal variables $(A, B)$ are sampled from the distribution described in Appendix C.3 (Equations (74) & (75)). These variables are then mapped to observations $(X, Y) \sim p_\mu(X, Y)$ via a hidden (and unknown to the learner) decoder $\mathcal{D} = R_{\theta_\mathcal{D}}$, where $R_\theta$ is a rotation of angle $\theta$. The observations are then mapped to the hidden state $(U, V) \sim p_\mu(U, V)$ via the encoder $\mathcal{E} = R_{\theta_\mathcal{E}}$; in this experiment, the angle $\theta_\mathcal{E}$ is the only additional meta-parameter, besides the structural parameter $\gamma$. The computational graph is depicted in Figure 3. In our experiment, $\theta_\mathcal{D} = -\pi/4$ is fixed for all our observation and intervention datasets. Interventional data is acquired by intervening on the latent variables $(A, B)$, following the process described in Appendix C.3, and then mapping the data through the decoder $\mathcal{D}$.

Since the underlying latent causal variables $(A, B)$ are unobserved, we need to define the online likelihood over the recovered variables $(U, V)$ instead. Analogous to how we defined the online likelihood in the fully observable case in Section 3, this is defined as

$$\mathcal{L}_G(\mathcal{D}_{int}\,;\,\theta_\mathcal{E}) = \prod_{t=1}^{T} p(R_{\theta_\mathcal{E}}(\mathbf{x}_t)\,;\,\theta_G^{(t)}, G) \qquad \begin{aligned} \theta_G^{(1)} &= \hat{\theta}_G^{ML}(R_{\theta_\mathcal{E}}(\mathcal{D}_{obs})) \\ \theta_G^{(t+1)} &= \theta_G^{(t)} + \alpha \nabla_\theta \log p(R_{\theta_\mathcal{E}}(\mathbf{x}_t)\,;\,\theta_G^{(t)}, G), \end{aligned} \tag{79}$$

where $R_{\theta_\mathcal{E}}(\mathcal{D}_{obs}) = \{R_{\theta_\mathcal{E}}(\mathbf{x}) \mid \mathbf{x} \in \mathcal{D}_{obs}\}$. Note that here the online likelihood depends on the parameters of the encoder $\mathcal{E}$ (here, $\theta_\mathcal{E}$). Using this definition of the online likelihood that takes into account the encoder, the meta-transfer objective is also similar to the one defined in Equation (5):

$$\mathcal{R}(\mathcal{D}_{int}\,;\,\gamma, \theta_\mathcal{E}) = -\log\left[\sigma(\gamma)\mathcal{L}_{U \to V}(\mathcal{D}_{int}\,;\,\theta_\mathcal{E}) + (1 - \sigma(\gamma))\mathcal{L}_{V \to U}(\mathcal{D}_{int}\,;\,\theta_\mathcal{E})\right]. \tag{80}$$

On the one hand, the gradient of $\mathcal{R}(\mathcal{D}_{int}\,;\,\gamma, \theta_\mathcal{E})$ with respect to the structural parameter $\gamma$ can be computed using Proposition 2, similar to the fully observable case. On the other hand, the gradient of the meta-transfer objective with respect to the meta-parameter $\theta_\mathcal{E}$ is computed using backpropagation through the $T$ updates of the parameters $\theta_G$ of the modules in Equation (79); this process is similar to backpropagation through time. In our experiment, we did not observe any degenerate behaviour like vanishing gradients, due to the limited amount of interventional data ($T = 5$).

## E  MORE THAN TWO CAUSAL HYPOTHESES

In Section 3.2, we defined the meta-transfer objective only in the context of bivariate models. The challenge with learning the structure of graphs on $n$ variables is that there is a super-exponential number of DAGs on $n$ variables, making the problem of structure learning NP-hard (Chickering, 2002a). If we were to naively extend the meta-transfer objective to graphs on $n > 2$ variables, this would require adaptation of $2^{O(n^2)}$ different models (hypotheses), which is intractable.

Instead, we can decouple the optimization of the graph from the acyclicity constraint, since causal graphs can have cycles (Peters et al., 2017). This constraint can be enforced as an extra penalty to the meta-transfer objective (Zheng et al., 2018). We consider the problem of optimization on the graph as $O(n^2)$ independent binary decisions on whether $V_j$ is a parent (or direct cause) of $V_i$. Motivated by the mechanism independence assumption (Parascandolo et al., 2017), we propose a heuristic to learn the causal graph, in which we independently parametrize the binary probability $p_{ij}$ that $V_j$ is a parent of $V_i$. We can then define a distribution over graphs (or more precisely, their adjacency matrix $B$) as:

$$B_{ij} \sim \text{Bernoulli}(p_{ij}) \tag{81}$$

$$p(B) = \prod_{i,j} p(B_{ij}), \tag{82}$$

where $p_{ij} = \sigma(\gamma_{ij})$. We denote $\text{Pa}_B(V_i)$ as the parent set of $V_i$ in the graph defined by the adjacency matrix $B$ (that is the nodes $V_j$ such that $B_{ij} = 1$). We can slightly rewrite the definition of the

online-likelihood as in Section 3 to show the dependence on $B$:

$$\mathcal{L}_B(\mathcal{D}_{int}) = \prod_{t=1}^{T} p(\mathbf{x}_t \,;\, \theta_B^{(t)}, B) = \prod_{i=1}^{n} \prod_{t=1}^{T} p(x_i^{(t)} \mid \mathbf{x}_{\text{Pa}_B(V_i)}^{(t)} \,;\, \theta_{B,i}^{(t)}), \tag{83}$$

where the second equality uses the factorization of $p$ in the graph defined by $B$. Note that since the graph defined by $B$ can contain cycles, the definition in Equation (83) involves the *pseudolikelihood* instead of the joint likelihood (which is defined as the product of individual conditional distributions only if the graph is a DAG). The pseudolikelihood was shown to be a reasonable approximation of the true joint likelihood when maximizing the joint likelihood (which is performed here for adaptation; Koller & Friedman (2009)). Similar to the bivariate case, we want to consider a mixture over all possible graph structures, but where each component must explain the whole adaptation sequence. We can generalize our definition of the regret as

$$\mathcal{R}(\mathcal{D}_{int}) = -\log \mathbb{E}_B[\mathcal{L}_B(\mathcal{D}_{int})]. \tag{84}$$

Note, however, that this expectation is over the $O(2^{n^2})$ possible values of $B$, which is intractable. We can rewrite the regret in a more convenient form:

**Proposition 4.** *The regret $\mathcal{R}(\mathcal{D}_{int})$ defined in Equation (84) can be decomposed as*

$$\mathcal{R}(\mathcal{D}_{int}) = -\sum_{i=1}^{n} \log \mathbb{E}_{B_i}[\mathcal{L}_{B_i}(\mathcal{D}_{int})], \tag{85}$$

*where $B_i$ is a row of the matrix $B$, and $\mathcal{L}_{B_i}(\mathcal{D}_{int})$ appears in the factorization of $\mathcal{L}_B(\mathcal{D}_{int})$ in Equation (83):*

$$\mathcal{L}_{B_i} = \prod_{t=1}^{T} p(x_i^{(t)} \mid \mathbf{x}_{\text{Pa}_B(V_i)}^{(t)} \,;\, \theta_{B,i}^{(t)}) \tag{86}$$

*Proof.* Recall that $\mathcal{L}_B(\mathcal{D}_{int}) = \prod_i \mathcal{L}_{B_i}(\mathcal{D}_{int})$, so that we can rewrite the regret as follows:

$$\mathcal{R}(\mathcal{D}_{int}) = -\log \mathbb{E}_B[\mathcal{L}_B(\mathcal{D}_{int})] \tag{87}$$

$$= -\log \sum_B p(B)\mathcal{L}_B(\mathcal{D}_{int}) \tag{88}$$

$$= -\log \sum_{B_1} \sum_{B_2} \cdots \sum_{B_n} \left[ \prod_{i=1}^{n} p(B_i)\mathcal{L}_{B_i}(\mathcal{D}_{int}) \right] \tag{89}$$

$$= -\log \prod_{i=1}^{n} \left[ \sum_{B_i} p(B_i)\mathcal{L}_{B_i}(\mathcal{D}_{int}) \right] \tag{90}$$

$$= -\sum_{i=1}^{n} \log \sum_{B_i} p(B_i)\mathcal{L}_{B_i}(\mathcal{D}_{int}) \tag{91}$$

$$= -\sum_{i=1}^{n} \log \mathbb{E}_{B_i}[\mathcal{L}_{B_i}(\mathcal{D}_{int})] \tag{92}$$

■

The structural parameters, here, are the $O(n^2)$ scalars $\gamma_{ij}$. Regardless of the intractability of the regret, we can still derive its gradient wrt. each $\gamma_{ij}$. The following proposition provides a direct extension of Proposition 2 to the case of multiple variables:

**Proposition 5.** *The gradient of the regret $\mathcal{R}(\mathcal{D}_{int})$ wrt. the structural parameter $\gamma_{ij}$ is given by*

$$\frac{\partial \mathcal{R}}{\partial \gamma_{ij}} = \sigma(\gamma_{ij}) - \sigma(\gamma_{ij} + \Delta_{ij}), \tag{93}$$

*where $\Delta_{ij}$ is the difference in log-likelihoods of two mixture candidates, conditioning on the variable $V_j$*

$$\Delta_{ij} = \log(\mathbb{E}_{B_i}[\mathcal{L}_{B_i}(\mathcal{D}_{int}) \mid V_j \in \text{Pa}_B(V_i)]) - \log(\mathbb{E}_{B_i}[\mathcal{L}_{B_i}(\mathcal{D}_{int}) \mid V_j \notin \text{Pa}_B(V_i)]) \tag{94}$$

*Proof.* To simplify the notation, we remove the explicit dependence on the transfer distribution $\mathcal{D}_{int}$ in this proof. Recall from Proposition 4 that the regret can be written as

$$\mathcal{R} = -\sum_{i=1}^{n} \log \mathbb{E}_{B_i}[\mathcal{L}_{B_i}]. \tag{95}$$

Using a conditional expectation, it follows that for any i, j

$$\mathbb{E}_{B_i}[\mathcal{L}_{B_i}] = \sum_{B_i} p(B_i)\mathcal{L}_{B_i} \tag{96}$$

$$= p(B_{ij} = 1) \cdot \sum_{B_i | V_j \in \mathrm{Pa}_B(V_i)} p(B_i \mid B_{ij} = 1)\mathcal{L}_{B_i} + p(B_{ij} = 0) \cdot \sum_{B_i | V_j \notin \mathrm{Pa}_B(V_i)} p(B_i \mid B_{ij} = 0)\mathcal{L}_{B_i} \tag{97}$$

$$= \sigma(\gamma_{ij}) \cdot \sum_{B_i | V_j \in \mathrm{Pa}_B(V_i)} p(B_i \mid B_{ij} = 1)\mathcal{L}_{B_i} + (1 - \sigma(\gamma_{ij})) \cdot \sum_{B_i | V_j \notin \mathrm{Pa}_B(V_i)} p(B_i \mid B_{ij} = 0)\mathcal{L}_{B_i} \tag{98}$$

To simplify the notation, let us define $E_{ij}^{(1)}$ and $E_{ij}^{(0)}$ the two conditional expectations of $\mathcal{L}_{B_i}$, conditioned on whether or not $V_j$ is a parent of $V_i$ in $B$

$$E_{ij}^{(0)} = \sum_{B_i | V_j \notin \mathrm{Pa}_B(V_i)} p(B_i \mid B_{ij} = 0)\mathcal{L}_{B_i} = \mathbb{E}_{B_i}[\mathcal{L}_{B_i} \mid V_j \notin \mathrm{Pa}_B(V_i)] \tag{99}$$

$$E_{ij}^{(1)} = \sum_{B_i | V_j \in \mathrm{Pa}_B(V_i)} p(B_i \mid B_{ij} = 1)\mathcal{L}_{B_i} = \mathbb{E}_{B_i}[\mathcal{L}_{B_i} \mid V_j \in \mathrm{Pa}_B(V_i)], \tag{100}$$

so that Equation (98) can be written as

$$\mathbb{E}_{B_i}[\mathcal{L}_{B_i}] = \sigma(\gamma_{ij})E_{ij}^{(1)} + (1 - \sigma(\gamma_{ij}))E_{ij}^{(0)}. \tag{101}$$

Note that neither $E_{ij}^{(0)}$ nor $E_{ij}^{(1)}$ depend on the structural parameter $\gamma_{ij}$. Therefore we can now easily compute the gradient of $\mathcal{R}$ wrt. $\gamma_{ij}$ only

$$\frac{\partial \mathcal{R}}{\partial \gamma_{ij}} = -\frac{\partial}{\partial \gamma_{ij}} \log \left( \sigma(\gamma_{ij})E_{ij}^{(1)} + (1 - \sigma(\gamma_{ij}))E_{ij}^{(0)} \right) \tag{102}$$

$$= -\frac{1}{\mathbb{E}_{B_i}[\mathcal{L}_{B_i}]} \left( \sigma(\gamma_{ij})(1 - \sigma(\gamma_{ij})) \left[ E_{ij}^{(1)} - E_{ij}^{(0)} \right] \right) \tag{103}$$

If we substract $\sigma(\gamma_{ij})$ from this expression gives us

$$\sigma(\gamma_{ij}) - \frac{\partial \mathcal{R}}{\partial \gamma_{ij}} = \frac{1}{\mathbb{E}_{B_i}[\mathcal{L}_{B_i}]} \left( \sigma(\gamma_{ij})^2 E_{ij}^{(1)} + \sigma(\gamma_{ij})(1 - \sigma(\gamma_{ij}))E_{ij}^{(0)} \right.$$
$$\left. + \sigma(\gamma_{ij})(1 - \sigma(\gamma_{ij})) \left[ E_{ij}^{(1)} - E_{ij}^{(0)} \right] \right) \tag{104}$$

$$= \frac{\sigma(\gamma_{ij})}{\mathbb{E}_{B_i}[\mathcal{L}_{B_i}]} E_{ij}^{(1)} = x \tag{105}$$

Denoting the previous expression as $x$, we can also easily compute $1 - x$:

$$1 - x = \frac{1 - \sigma(\gamma_{ij})}{\mathbb{E}_{B_i}[\mathcal{L}_{B_i}]} E_{ij}^{(0)} \tag{106}$$

Using the logit function $\sigma^{-1}(x) = \log \frac{x}{1-x}$, we can conclude that

$$\sigma^{-1}\left( \sigma(\gamma_{ij}) - \frac{\partial \mathcal{R}}{\partial \gamma_{ij}} \right) = \log \frac{\sigma(\gamma_{ij})E_{ij}^{(1)}}{(1 - \sigma(\gamma_{ij}))E_{ij}^{(0)}} \tag{107}$$

$$= \log \frac{\sigma(\gamma_{ij})}{1 - \sigma(\gamma_{ij})} + \log E_{ij}^{(1)} - \log E_{ij}^{(0)} \tag{108}$$

$$= \gamma_{ij} + \Delta_{ij} \tag{109}$$

$$\blacksquare$$

While Proposition 5 gives an analytic form for the gradient of the regret wrt. the structural parameters, computing it is still intractable, due to $\Delta_{ij}$. However, we can still get an effecient stochastic gradient estimator from Proposition 4, which can be computed separately for each node of the graph (with samples arising only out of $B_i$, the incoming edges of $V_i$):

**Proposition 6.** *If we consider multiple samples of $B$ in parallel, a biased but asymptotically unbiased (as the number $K$ of these samples $B^{(k)}$ increases to infinity) estimator of the gradient of the overall regret with respect to the meta parameters can be defined as:*

$$g_{ij} = \frac{\sum_k (\sigma(\gamma_{ij}) - B_{ij}^{(k)}) \mathcal{L}_{B_i}^{(k)}}{\sum_k \mathcal{L}_{B_i}^{(k)}}, \tag{110}$$

*where the index $^{(k)}$ indicates the values obtained for the $k$-th draw of $B$.*

*Proof.* The gradient of the regret with respect to the meta-parameters $\gamma_i$ of node $i$ is

$$\frac{\partial \mathcal{R}}{\partial \gamma_i} = -\frac{\sum_{B_i} p(B_i) \mathcal{L}_{B_i} \frac{\partial \log p(B_i)}{\partial \gamma_i}}{\sum_{B_i} P(B_i) \mathcal{L}_{B_i}}$$

$$= -\frac{\mathbb{E}_{B_i}[\mathcal{L}_{B_i} \frac{\partial \log p(B_i)}{\partial \gamma_i}]}{\mathbb{E}_{B_i}[\mathcal{L}_{B_i}]} \tag{111}$$

Note that with the sigmoidal parametrization of $p(B_i)$,

$$\log p(B_i) = B_{ij} \log \sigma(\gamma_{ij}) + (1 - B_{ij}) \log(1 - \sigma(\gamma_{ij})) \tag{112}$$

as in the cross-entropy loss. Its gradient can similarly be simplified to

$$\frac{\partial \log p(B_{ij})}{\partial \gamma_{ij}} = \frac{B_{ij}}{\sigma(\gamma_{ij})} \sigma(\gamma_{ij})(1 - \sigma(\gamma_{ij})) - \frac{(1 - B_{ij})}{(1 - \sigma(\gamma_{ij}))} \sigma(\gamma_{ij})(1 - \sigma(\gamma_{ij})))$$

$$= B_{ij} - \sigma(\gamma_{ij}). \tag{113}$$

A biased, but asymptotically unbiased, estimator of $\partial \mathcal{R} / \partial \gamma_{ij}$ is thus obtained by sampling $K$ graphs (over which the means below are run):

$$g_{ij} = \sum_k (\sigma(\gamma_{ij}) - B_{ij}^{(k)}) \frac{\mathcal{L}_{B_i}^{(k)}}{\sum_{k'} \mathcal{L}_{B_i}^{(k')}} \tag{114}$$

where index $^{(k)}$ indicates the $k$-th draw of $B$, and we obtain a weighted sum of the individual binomial gradients weighted by the relative regret of each draw $B_i^{(k)}$ of $B_i$, leading to Equation (110). ∎

We can therefore adapt Algorithm 1 using the gradient estimate in Proposition 6 to update the structural parameters $\gamma_{ij}$, without having to explicitly compute the full regret $\mathcal{R}(\mathcal{D}_{int})$. In addition to the gradient estimate provided by Proposition 6, we can also derive a Rao-Blackwellized (Rao, 1992; Blackwell, 1947) estimate of the gradient of the regret, based on the formulation derived in Proposition 5.

**Proposition 7.** *Let $\{B^{(k)}\}_{k=1}^K$ be $K$ binary matrices (corresponding to sample graphs), sampled from independent Bernoulli distributions depending on the structural parameters $\gamma_{ij}$*

$$B_{ij}^{(k)} \overset{iid}{\sim} \text{Bernoulli}(\sigma(\gamma_{ij})), \tag{115}$$

*and their corresponding likelihoods $\mathcal{L}_{B_i}^{(k)}$. A Monte-Carlo estimate of the log-likelihood difference $\Delta_{ij}$ in Equation (94) is given by*

$$\widetilde{\Delta}_{ij}^{(K)} = \log\left(\frac{1}{|\mathcal{K}_{ij}^{(1)}|} \sum_{k \in \mathcal{K}_{ij}^{(1)}} \mathcal{L}_{B_i}^{(k)}\right) - \log\left(\frac{1}{|\mathcal{K}_{ij}^{(0)}|} \sum_{k \in \mathcal{K}_{ij}^{(0)}} \mathcal{L}_{B_i}^{(k)}\right), \tag{116}$$

*where $\mathcal{K}_{ij}^{(0)} = \{k \,;\, B_{ij}^{(k)} = 0\}$ and $\mathcal{K}_{ij}^{(1)} = \{k \,;\, B_{ij}^{(k)} = 1\}$ are (disjoint) sets of indices $k$, depending on the value of $B_{ij}^{(k)}$.*

*Based on this Monte-Carlo estimate of $\Delta_{ij}$, we can define an estimate of the gradient of the regret $\mathcal{R}$ wrt. the structural parameter $\gamma_{ij}$ by*

$$\frac{\widetilde{\partial \mathcal{R}}}{\partial \gamma_{ij}} = \sigma(\gamma_{ij}) - \sigma(\gamma_{ij} + \widetilde{\Delta}_{ij}^{(K)}). \tag{117}$$

## F  THE EFFECT OF ADAPTATION ON THE ONLINE LIKELIHOOD

Since we are using the online likelihood defined in Equation (3) as a measure of adaptation in our meta-transfer objective, it is reasonable to know if this measure is sound. To validate this assumption, we are running an experiment similar to the one described in Section 2.1 and Figure 1, using the same experimental setup on discrete variables described in Section 3.3. However, instead of measuring the raw log likelihood on a validation set, we report the online likelihood $\mathcal{L}_G(\mathcal{D}_{int})$ for both models in Figure F.1. The online likelihoods are scaled by the number of transfer examples seen for visualization. Similar to Figure 1, we can see that the difference in online likelihoods for both models is most significant on a small amount of data.

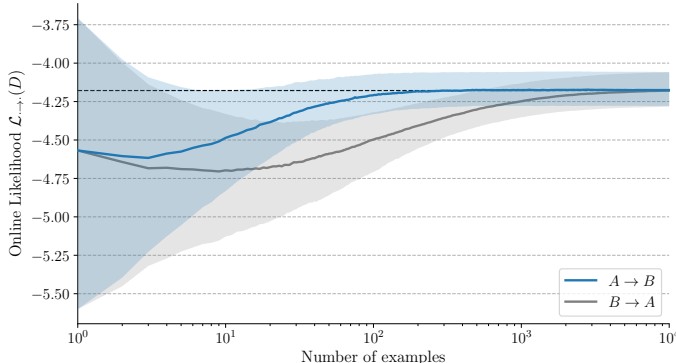

Figure F.1: Adaptation to the transfer distribution (online likelihood on transfer examples, vertical axis), as more transfer examples are seen by the learner (horizontal axis). The curves are the median over 20,000 runs, with their 25-75th quantiles intervals. The dotted line is the asymptotic online likelihood

