# OpenReview forum: "A Meta-Transfer Objective for Learning to Disentangle Causal Mechanisms"
_ICLR.cc/2020/Conference — Accept (Poster)_

### Official Review · AnonReviewer2 · 2019-10-18
**Official Blind Review #2**

**Rating:** 8

**Review:**


Summary:
The paper first shows that, in a very simple two-variable task, the model with the correct underlying structure will adapt faster to a causal intervention than the model with the incorrect structure. This idea is used to develop a “meta-transfer” objective function for which gradient ascent on a continuous representation of the model structure allows learning of that structure. The paper shows that optimizing with respect to this objective with a simple model is guaranteed to converge to the correct structure, and also presents experimental results on toy problems to demonstrate.

Overall: Accept.
I really enjoyed reading this paper. It is clear, well-motivated, well-written, does a good job of connecting to related work, and presents an interesting method for structure learning. While the experiments are quite toy and questions about how well this will work in more complex models with many variables remain largely unaddressed, these do not detract much from the paper for me. Instead, the paper does a good job of motivating its contribution and exploring its effect in simple intelligible tasks, and I feel I got more out of this paper than most SOTA papers.


Clarity: Very clear.
Significance: Potentially quite significant as this is starting to bring causal structure learning into the realm of tensorflow and pytorch.

Questions and comments:
- All else being equal, the speed of adaptation between two very similar models will serve as a good proxy, as shown in this paper. However, I can easily imagine scenarios where the two models one wants to differentiate between are quite different, and have very different optimization landscapes. Here, the speed of adaptation will be quite dependent on these landscapes and not just on the underlying model structure. Do you have thoughts about how this can be extended to such a scenario?

- The parameter counting argument is not nearly so strong if what actually changes is the conditional p(A|B). In that case, the sample complexity for the correct model would be N^2 = O(N^2) and for the incorrect model would be N + N^2 = O(N^2). Does the objective still work here? Would be great to add an additional experiment showing the results in this case.

- Doing an intervention and drawing a new D_int for each step of gradient descent seems quite prohibitive in a lot of domains. Are there ways to decrease this burden?

- In Figure 2, can you speak to why the N=100 curve for the MLP parameterization converges more slowly than the N=10 curve? I would still expect more data to be beneficial here.


**Experience Assessment:**

I have published one or two papers in this area.

**Review Assessment: Checking Correctness Of Derivations And Theory:**

I assessed the sensibility of the derivations and theory.

**Review Assessment: Checking Correctness Of Experiments:**

I carefully checked the experiments.

**Review Assessment: Thoroughness In Paper Reading:**

I read the paper at least twice and used my best judgement in assessing the paper.

---

> ### Author Response · Authors · 2019-11-13
> **Response to Reviewer 2**
>
> We would like to thank you for your kind words about the paper, and we share your enthusiasm regarding this research direction. We want to answer your questions here:
>
> - In our experiments, we carefully made sure that the two models A -> B and B -> A have the same capacity, as shown in Tables D.1-4, to ensure that there was no bias induced by the modeling choice we could not control for. However it is indeed very interesting to ask what would happen if this assumption does not hold. There are several factors that can contribute to the “fast adaptation”, but in general one would expect that models more faithful to reality lead to faster adaptation; causality is one such aspect that we are investigating in this paper, but other aspects might exist as well. Nonetheless, as is the case in Machine Learning in general, a good but difficult to optimize model may be rejected over a model which is easier to train from an optimization point of view. In that case, it might be impossible to recover the causal direction alone, independently of all other factors of variations (such as different model structures). As an extreme example, even the loss landscape involved in the computation of the online likelihood (Equation (3)) could have an impact on the conclusions about the causal directions; we did not encounter this issue in our experiments though.
>
> - Following this suggestion, we included an experiment where the conditional distribution p(B | A) changes in Appendix D.5 in the first revision. In the absence of the parameter counting argument, we found in our initial experiments that the conclusions may not hold anymore. To summarize our findings so far: our method sometimes fails and sometimes works (albeit with a measure of adaptation performance different from the online likelihood used throughout the paper). We would also like to note that even though the last part of Section 2.2 does not hold, the result of Proposition 1 still applies in this situation. The conclusion of the paper remains that when the counting argument works (which here means an intervention on the cause), our experiments show successful recovery of the cause-effect relationship.
>
> - Drawing a new $D_{int}$ for each step of gradient descent might seem prohibitive. However, the purpose of the meta-transfer objective is to leverage information from only small datasets $D_{int}$, meaning that there is not much of a burden when it comes to the amount of data required, using as little as T=20 datapoints for each intervention in our experiments. This burden is comparable to the one required by any meta-learning algorithm in general.
>
>  - Some clarification may be needed on the meaning of N. N here refers to the number of possible values of the categorical variables A and B can take, as mentioned at the end of Section 2.2 and in Section 3.3, and not to the amount of data (denoted in the paper as either m for $D_{obs}$ or T for $D_{int}$). A larger value of N puts the learning further in the “small-sample regime” where we have seen that the causal signal is stronger in Section 2.2 (and shown clearly in the tabular case in Figure 2, left). However, the difference is less clear for MLPs, most likely because unlike the tabular representation, the number of parameters in the MLP representation does not scale quadratically with N anymore, and MLPs are known to be robust to over-parametrization (i.e., they don't overfit easily even when the number of parameters is much larger than the number of examples).

---

> > ### Comment · AnonReviewer2 · 2019-11-15
> > **Acknowledgment**
> >
> > Thank you for the clarifications and updated paper, and I applaud the inclusion of a negative result in the Appendix for the p(B|A) scenario. My score remains unchanged.

---

### Official Review · AnonReviewer3 · 2019-10-30
**Official Blind Review #3**

**Rating:** 8

**Review:**

In this work, the authors proposed a general and systematic framework of meta-transfer objective incorporating the causal structure learning under unknown interventions. Under the assumption of small change (out-of-distribution data), the work mainly focuses on the theoretical and empirical analysis of relations on two random variables in causal graphs (causal and anti-causal directions), so that a differentiable regret function using the joint distribution of the small "intervention" dataset can be built.

The motivation is to adapt or transfer quickly by discovering the correct causal direction and learning representation based on it. The idea of disentangling the marginal and conditional factors to reduce the sample complexity and thus achieve fast adaptation is novel and insightful. Proposition 1 and its proof provide the theoretical supports on this point very well. The structure causal model is parametrized and then optimized in a meta-learning procedure. Experiments on simulated data under categorical or continuous distributions can verify the efficiency of inferring causal graphs.

Here are some concerns about the proposed algorithm:

1). When the authors discussed small change, there is no formal (mathematical) definition on it. For instance, an invertible function could be one of the properties given for the out-of-distribution data. The example (rotation) in Fig. 3 works to some extent because the small transformation is invertible. Also, the intervention seems simple in the work, for example, the rotate angle (a value) in Fig. 3 only involves one parameter dimension. In this case, learning an encoder to infer the correct causal relation is not that difficult. Is there possible that the encoder cannot learn a good enough theta to find the correct causal direction? It would be nice if the limitations of using causal graphs are discussed.

2). Given a direction A causes B, the experiments are conducted by performing interventions on the cause A. How about to put an intervention on the effect B? According to the algorithm analysis (Table D.1), for the discrete bivariate model, the parameter dimension of a correct structure becomes N^2, while the one of an incorrect structure becomes N + N^2. Compared to intervention on cause, the reduction of sample complexity here is not that obvious. A general discussion on the effect intervention for bivariate models would be helpful.

3). The work opens a new direction of inferring causal relationships together with representation learning, which has the potential for more out-of-distribution scenarios. While the authors claim that it is the first step, the current empirical studies for structure models use synthetic data with relatively constraint assumptions. It is highly recommended for the authors to provide discussions about real-data tasks with neural causal models in future work.


**Experience Assessment:**

I have read many papers in this area.

**Review Assessment: Checking Correctness Of Derivations And Theory:**

I carefully checked the derivations and theory.

**Review Assessment: Checking Correctness Of Experiments:**

I carefully checked the experiments.

**Review Assessment: Thoroughness In Paper Reading:**

I read the paper thoroughly.

---

> ### Author Response · Authors · 2019-11-13
> **Response to Reviewer 3**
>
> We would like to thank you for your interest in the paper, and for carefully going through our theory and experimental results. We want to address your three concerns here in more detail:
>
> 1) There might be a small confusion here about our assumption of “small changes”. This assumption applies to the intervention on the distribution p (throughout the paper) over causal variables, and not to the encoder (which is in Section 4 only). More precisely, we assume that the transfer distribution $\tilde{p}$ is the result of an intervention on only a single variable (in the paper, the cause A); the fact that we only intervene on a single variable characterizes the “localized” (or “small”) nature of the change. This assumption is detailed in Section 2.1, and mathematically defined in Proposition 1. This change is small only in the right representation space (e.g. in the latent space in our experiment in Section 4), as mentioned in Section 1. On the other hand, we do not make any assumption on the magnitude of the angle for the decoder; in our experiment, we used $-\pi/4$ (see Appendix E in the first revision for details).
>
> 2) Following the suggestion of the reviewer, we conducted an experiment where the conditional distribution p(B | A) changes while the marginal on the cause p(A) remains unchanged during intervention. Our experiment results are available in Appendix D.5 in the first revision. Moreover, even if the change is on p(B | A), our analysis of the number of dimensions for each model in Table D.1 would still be valid. The learner does not know that p(A) is unchanged, and thus we would still have to model the marginal of A, similar to how we still had to model the conditional of B given A in the experiment of Appendix D.1. This ensures that both models have the same capacity (inducing any spurious bias). That being said, as pointed out by Reviewer 2, the parameter counting argument at the end of Section 2.2 would not hold if the intervention is on the effect B.
>
> 3) We share the sentiment that future work should include discussions on the application of the meta-transfer objective to real-data tasks. Following up on this work, [1] showed empirical success on both graphs with multiple variables, as well as standard datasets from the Bayesian Network Repository [2].
>
> [1] Ke, Nan et al., Learning Neural Causal Models from Unknown Interventions (2019).
> [2] Bayesian Network Repository: http://www.bnlearn.com/bnrepository/.

---

### Official Review · AnonReviewer5 · 2019-11-01
**Official Blind Review #5**

**Rating:** 3

**Review:**

The paper proposes a method of discovering causal mechanisms through meta-learning, by assuming that models transfer faster if their causal graph is correct.

For a possible causal graph, for adaptation to one interventional dataset, the samples are iteratively revealed and the log likelihood of the next sample is measured, after which the parameters are updated using one step of gradient descent on that sample. The sum of these log likelihoods is the ‘online likelihood’ and a measure of speed of adaptation. The parameters are initialised using maximum likelihood estimation on the train dataset.
The meta learning procedure then at each episode samples an interventional distribution and an interventional dataset from that distribution. It then performs a gradient based update of the belief over graphs based on the difference between the online likelihoods of each graph on that dataset.

The meta-learning approach appears to be a novel contribution. The authors provide a theoretical argument by counting ‘effective parameters’ to suggest why models using the right causal model obtain a higher online likelihood. Additionally, they prove that the gradient updates to the graph belief are easy to compute and converge. The method is validated with several synthetic experiments which discover the direction of the arrow between two random variables, each either continuous or discrete. Furthermore, they successfully experiment with the combination of learning a representation of a raw data to two random variables with learning the causal direction. The paper is very well written and most claims are carefully proven.

I recommend a weak rejection for this paper, because:
1) The empirical validation is not strong enough, as no real dataset is used, only toy datasets. The toy experiments themselves could also be more extensive.
2) I am unconvinced of two of the theoretical claims made: (A) the fact that the expected gradients in Prop 1 are 0, implies that the right causal graph has better online likelihood and (B) that the method is easily extensible to more than two random variables [Appendix E].

Supporting arguments:
1.1) Fig D.1 suggests that, in the experiments using continuous random variables, the training dataset alone is sufficient to discover the true causal model, under the assumption of independent additive noise, as is done in e.g. [1]. I find it plausible to believe that the training curve on the training dataset alone already makes it possible to disambiguate the causal from anti-causal model. A similar pattern is shown by the authors themselves in Appendix B on discrete variables.
For finite training data, the models are distinguishable, while for infinite training data they are not [Fig B.1]. The exact same holds for finite and infinite interventional data [Fig 1].
Are these experiments then good benchmarks for causal discovery based on intervention when the causal model can already be inferred from the non-interventional training data?
1.2) The simplicity of the representation learning setup doesn’t convince that the method is applicable to more real-world settings with more complicated encoders. Additionally, some important details on this experiment are missing (see below).
1.3) All experiments show the effect of intervention on the cause p(A). No experiments are given for intervention on p(B|A). Does the method then still work?
1.4) No experiments for more than two random variables are performed in this paper.
2.A) Prop 1 shows that the expected maximum likelihood gradient for one conditional probability distribution is zero if the graph is correct and that CPD is not intervened on. Subsequently a claim is made that this effectively reduces the number of parameters and thus that adaptation is faster. However, the zero-expectation gradient may still be non-zero and even large on the small intervention sample. It is unclear to me why they therefore can be excluded in the number of parameters. Furthermore, whether the online likelihood is large will depend not only on generalisation, but also on the training convergence, since not empirical risk minimization is used, but SGD with a fixed number of steps. Thus, even though the authors prove that the method will converge to the causal graph with lowest online likelihood, it is unclear why this is necessarily the correct causal graph.
2.B) In appendix E it is mentioned that cycles can occur in causal models. However, it is unclear why factorization (76) is still correct in the cyclic case. Perhaps I am misunderstanding, but it seems to me that p(x) = \prod_i p(x_i | x_{Pa_i}) only makes sense for a DAG. Hence, how would the online likelihood be computed for cyclic models?

Suggestions for improvement:
- Could you explain in what realistic settings we would have access to data from a large number of different interventional distributions?
- Could you show a plot similar to Fig 1, but with online likelihoods? Such a plot may be more indicative of the ideal episode length than Fig 1.
- Could you provide details on the representation learning experiment? In particular: (1) is \theta_D different in the train dataset and each interventional dataset? (2) How do the gradients of theta flow through the meta-update steps?
- A reference to Appendix B appears missing in main text.

[1] Nonlinear causal discovery with additive noise models. Hoyer et al 2009

**Experience Assessment:**

I have published one or two papers in this area.

**Review Assessment: Checking Correctness Of Derivations And Theory:**

I assessed the sensibility of the derivations and theory.

**Review Assessment: Checking Correctness Of Experiments:**

I carefully checked the experiments.

**Review Assessment: Thoroughness In Paper Reading:**

I read the paper thoroughly.

---

> ### Author Response · Authors · 2019-11-13
> **Response to Reviewer 5**
>
> We would like to thank you for the time you have invested in reviewing our work. We are glad that you find our contribution novel and our paper well written, and your suggestions have helped us significantly improve the paper. Our detailed response to your comments follows.
>
> 1.1)
> > Are these experiments then good benchmarks for causal discovery based on intervention when the causal model can already be inferred from the non-interventional training data?
>
> This is indeed a valid point, and we provide an additional supporting experiment where causal discovery fails for non-interventional training data (a linear-gaussian model, which is not identifiable), but succeeds with interventional data. This can be found in Appendix D.3.
>
> 1.2)
> > The simplicity of the representation learning setup doesn’t convince that the method is applicable to more real-world settings with more complicated encoders. Additionally, some important details on this experiment are missing (see below).
>
> We agree with your assessment that this is only a first step -- a proof of concept in a minimal setting -- and that much more work is required to explore and understand this very important, but challenging, direction. More complex experimental settings with the encoder are therefore left for future work.
>
> Regarding the missing important details -- thank you for pointing this out. We have added them in Appendix E of the revision to address this.
>
> 1.3)
> > All experiments show the effect of intervention on the cause p(A). No experiments are given for intervention on p(B|A). Does the method then still work?
>
> Thank you for the important suggestion. We have good reasons to believe that the cause intervention will generally work best based on the parameter counting argument. We have conducted additional experiments in a setting where the conditional distribution p(B|A) changes while the marginal distribution p(A) remains unchanged, and the results are indeed mixed. To summarize our findings so far: in some cases our method did not work, in others our method can work, with a measure of adaptation performance different from the online likelihood used throughout the paper. The results, together with one failure case, can be found in Appendix D.5.
>
> 1.4)
> > No experiments for more than two random variables are performed in this paper.
>
> The focus of the current work is placed on the important class of bivariate causal graphs (cf. chapters 1-5 of Peters et al. [1]). Our objective is to introduce the meta-transfer objective in the two-variable case, thereby laying down the foundations for future work combining causal structure learning with deep learning on larger graphs. While we include theoretical results in Appendix E (Appendix F in the revision), further results on general multivariate graphs is left to future work by the community. For instance, more recent work [2] builds on our insights to show positive experimental results on graphs with up to 8 variables.
>
> 2.A)
> > However, the zero-expectation gradient may still be non-zero and even large on the small intervention sample. It is unclear to me why they therefore can be excluded in the number of parameters.
>
> While the adaptation is performed on the small intervention dataset $D_{int}$, the structural parameter $\gamma$ is updated by SGD over many intervention distributions $\tilde{p}$, justifying our result on expectation over $\tilde{p}$ in Proposition 1. On average over meta-training, the parameter gradients of the unmodified modules will be zero, having a smaller contribution during the computation of the online likelihood for the correct causal model. We think that it would be possible to exploit the kind of theoretical analysis which has been made for sparse regression (where only a few of the inputs need to have non-zero weights, while the expected gradients on the weights from the other inputs would be zero if those weights are set to zero). In that case it can be shown [3] that the capacity scales linearly with the number of truly dependent variables (the number of non-zero true weights), and thus the number examples needed also only need to scale with that number.
>
> (continuing in next comment)

---

> > ### Author Response · Authors · 2019-11-13
> > **Response to Reviewer 5 (2)**
> >
> > (continuing from previous comment)
> >
> > 2.B)
> > > Perhaps I am misunderstanding, but it seems to me that p(x) = \prod_i p(x_i | x_{Pa_i}) only makes sense for a DAG. Hence, how would the online likelihood be computed for cyclic models?
> >
> > It is true that the factorization in the definition of the online likelihood only makes sense if the candidate graph contains no cycle. To be more precise, if the graph contains a cycle, we can use the pseudolikelihood in place of the likelihood here (leaving the definition unchanged). Although the pseudolikelihood is only an approximation of the joint likelihood, it has been shown to be a reasonable estimate for maximization (which is performed here for adaptation), instead of maximizing the joint likelihood [4]. We have updated Appendix E (Appendix F in the revision) to clarify this. Also note that what we have is a weighted form of log-pseudolikelihood (with each log-prob weighted by the edge probabilities) so that when learning converges (with appropriate regularization) these edge probabilities define a DAG and the weighted log-pseudolikelihood becomes a proper log-likelihood.
> >
> > We would like to also address your suggestions:
> >
> > > Could you explain in what realistic settings we would have access to data from a large number of different interventional distributions?
> >
> > The setting described here is similar to the few-shot episodic setting frequently encountered in meta-learning, where small datasets (here $D_{int}$) of different tasks (here different interventions) are used for meta-training. We expect these kinds of settings to become especially relevant in RL and multi-agent scenarios (where agents' actions are interventions).
> >
> > > Could you show a plot similar to Fig 1, but with online likelihoods? Such a plot may be more indicative of the ideal episode length than Fig 1.
> >
> > Indeed -- please refer to Figure G.1 in Appendix G in the revision.
> >
> > > Could you provide details on the representation learning experiment? In particular: (1) is \theta_D different in the train dataset and each interventional dataset? (2) How do the gradients of theta flow through the meta-update steps?
> >
> > We have included Appendix E in the first revision to provide experimental details. To answer your questions:
> > (1): \theta_D is constant throughout.
> > (2): The gradients to theta flow through the optimization process of the transfer episode (similar to MAML [5]).
> >
> > > A reference to Appendix B appears missing in main text.
> >
> > This has been fixed -- thank you for the pointer!
> >
> > In conclusion: thank you again for your detailed review. We hope our response and the revision addresses your concerns; please feel invited to respond to our comment if anything remains unclear.
> >
> > [1] Peters, Jonas et al., Elements of Causal Inference.
> > [2] Ke, Nan et al., Learning Neural Causal Models from Unknown Interventions (2019).
> > [3] Ng, Andrew, On feature selection: learning with exponentially many irrelevant features as training examples (1998).
> > [4] Koller, Daphne and Friedman, Nir, Probabilistic Graphical Models: Principles and Techniques.
> > [5] Finn, Chelsea et al., Model-Agnostic Meta Learning for Fast Adaptation of Deep Networks (2017).

---

> > > ### Comment · AnonReviewer5 · 2019-11-13
> > > **Further questions regarding parameter counting**
> > >
> > > Thank you for the extensive comment and the additions to the paper. I have some remaining questions regarding your key hypothesis: fewer non-zero expected parameter gradients implies better online likelihood.
> > >
> > > 1) I don’t quite follow the statement in your comment “On average over meta-training, the parameter gradients of the unmodified modules will be zero, having a smaller contribution during the computation of the online likelihood for the correct causal model.” What you average during the meta-train iterations is not the gradients to the online likelihood, but the online likelihood itself. Could you please clarify this?
> > >
> > > 2) Your “failure case” also shows that the parameter counting argument does not always imply better online likelihood (for finite N) and that the opposite can be true also. In this case, the right causal model with N^2 nonzero expected gradients has lower expected online likelihood than the wrong causal model with N^2+N nonzero expected gradients. Do you have an hypothesis why this is the case?
> > >
> > > 3) As you don’t transfer until convergence on the interventional training set (Fig 1), I find it plausible that for complicated models the training dynamics might affect the online likelihood in sufficiently to change the ordering of the causal graphs. Could you elaborate on this?
> > >
> > > Furthermore: could you please explain why in the new Figure D.4 without interventions, the line still appears to tend significantly towards the right causal model?

---

> > > > ### Author Response · Authors · 2019-11-14
> > > > **Response to the questions on parameter counting**
> > > >
> > > > 1) It is true that the online likelihood itself is averaged during meta-training and not the gradient of the online likelihood. Nonetheless, the online likelihoods are computed using parameters that have been updated using stochastic gradient descent. Adapting on only a small dataset of intervention data indeed means that we might not reach zero gradient, but we will have a noisy estimate of this gradient, which in turn might induce some noise in our measure of adaptation. The fact that we can repeat this multiple times on different transfer distributions over the course of meta-training means that the effect of this noise is mitigated. In practice, we did observe that the gradient of the meta-transfer objective with respect to the structural parameters (which is driven by the difference in online likelihoods, from Proposition 2) could sometimes point in the “wrong direction” (as is the case in SGD in general), but points on average in the direction leading to convergence towards the correct causal model.
> > > >
> > > > 2) We believe that the effect of the parameter counting argument depends more on the difference in the order of magnitude between the two causal models (10 vs 110 in the case of N=10 and p(A) changing, as given by Table D.1), rather than actual counts (100 vs 110 for N=10 and p(B | A) changing). This would explain why its effect is stronger using the tabular representation as N increases (Figure 2, left), and why the difference is less clear when increasing N in the experiment with MLP representation (Figure 2, right).
> > > >
> > > > 3) In all experiments, we made sure to control for as many sources of spurious bias (which would steer the decision towards one of the two hypotheses) as possible, notably choosing models with the same capacity to parameterize both causal models. It is true that even optimization could affect how the online likelihood is computed, and therefore induce enough bias to have the structure converge towards the wrong causal model. However across all our experiments with a interventions on the cause A, we did not encounter this situation on both simple models (eg. tabular representation for discrete variables) and more complex models (Mixture Density Networks for continuous multimodal variables). We also addressed this in part in the first point of our initial response to Reviewer 2, for an additional perspective.
> > > >
> > > > Finally we have updated Figure D.4 in our latest revision of the paper. The learning curves in the first revision were only averaged over 5 runs, where the line without intervention (in orange) was indeed leaning towards the right causal model. In the latest revision of the paper, we now average both curves over 40 runs, with a clearer conclusion: on average, the belief that A -> B is the correct causal model remains at 0.5. And even though some runs deviate from an equal belief (likely due to noise induced by SGD), using interventions proves to be more reliable in this case where the model is not identifiable.

---

### Author Response · Authors · 2019-11-13
**General response to all reviewers**

We would like to thank the three reviewers for their insightful comments. We really appreciated the feedback and suggestions to improve the paper. We would like to address here some points relevant to all three reviews, and add details for each individual reviewer in separate comments.

In this paper, we only considered the case where the true causal graph has two nodes (i.e. the cause-effect setting). This both makes the presentation clearer, and allows us to lay the foundations for general causal graphs. Although we discuss possible extensions to multiple variables in Appendix E (Appendix F in the first revision), we would like to stress that extensive experimentations on general causal are left for future work. We included this opening to multiple variables in the Appendix, as part of our initial release of this paper, to encourage other researchers to build on the ideas presented in the paper. For example, [1] builds on from our initial paper and showed empirical success on graphs with multiple variables using the formulation presented in this Appendix; we referenced this work in the ICLR version of the paper.

While we chose to focus our attention on a setting where we could intervene on the cause to get a stronger intuition, some reviewers are rightfully asking if the proposed ideas still hold if p(B | A) changes instead of p(A). In that case, our parameter counting argument from Section 2.2 would indeed not hold. Following the suggestion of the reviewers, we experimented with settings where the interventions are the results of changes on p(B | A) (leaving p(A) invariant, meaning that our change in distribution is still localised). Our initial experiments are mixed, depending on the experimental setup, which reinforces our argument based on parameter counting. These new results will be included in Appendix D.5 of the revision (together with one failure case), but would require further investigation. However, these remarks from the reviewers we will also help us clarify in the paper that we only claim that causal graphs have been recovered in the case where it is the cause which has been the subject of intervention, because this is the case where the counting argument applies.

We have uploaded a first revision of the paper with the following updates:
 - We added a missing reference to Appendix B in Section 2, as mentioned by Reviewer 5.
 - We have updated the proof of Proposition 1 (in Appendix C.1) which had a small mistake making it incorrect. We would like to thank a researcher that privately reached out to us about this proof.
- We have included results for changes on p(B | A) in Appendix D.5, as suggested by the reviewers.
- We have added a figure similar to Figure 1 with the online likelihood instead of the likelihood on an external validation set in Appendix G, as suggested by Reviewer 5.
- We have added a sanity check in Appendix D.3 where we apply our continuous bivariate setting to an SCM which is known to be not identifiable from a single distribution, as requested by Reviewer 5.
- We have added the Appendix E, containing details about our experiment on Representation Learning in Section 4. In particular, we mention how the decoder is fixed for all datasets, and we are detailing how the gradient with respect to the encoder’s parameters is computed as asked by Reviewer 5.
- We have clarified the use of a form of pseudolikelihood (rather than likelihood) in the definition of the online likelihood in Appendix E (Appendix F in the revision), as suggested by Reviewer 5.

[1] Ke, Nan et al., Learning Neural Causal Models from Unknown Interventions (2019).

---

### Decision · Program_Chairs · 2019-12-19

**Decision:**

Accept (Poster)

**Comment:**

This paper proposes to discover causal mechanisms through meta-learning, and suggests an approach for doing so. The reviewers raised concerns about the key hypothesis (that the right causal model implies higher expected online likelihood) not being sufficiently backed up through theory or through experiments on real data. The authors pointed to a recent paper that builds upon this work and tests on a more realistic problem setting. However, the newer paper measures not the online likelihood of adaptation, but just the training error during adaptation, suggesting that the approach in this paper may be worse. Despite the concerns, the reviewers generally agreed that the paper included novel and interesting ideas, and addressed a number of the reviewers' other concerns about the clarity, references, and experiments. Hence, it makes a worthwhile contribution to ICLR.